# A modelling system for identification of maize ideotypes, optimal sowing dates and nitrogen fertilization under climate change – PREPCLIM-v1

Mihaela Caian[1,] Catalin Lazar[2], Petru Neague[1], Antoanela Dobre[1], Vlad Amihaesei[1], Zenaida Chitu[1],
Adrian Irasoc[1], Andreea Popescu [1], George Cizmas[2]

[1] National Meteorological Administration Romania (NAM), Sos. Bucureşti-Ploieşti nr.97, Sector 1, 013686 Bucureşti România

[2] National Agricultural Research and Development Institute (NARDI) Fundulea, 915200 Călăraşi, România

*Correspondence to*: Mihaela Caian (mihaela.caian@gmail.com)

**Abstract.** Climate change significantly threatens crop yields levels and stability. The complex interplay of factors at the local scale makes assessing these impacts difficult, requiring coupled climate-phenology models, which integrate climate data and crop information. Identifying suitable local management practices and crop varieties under future conditions becomes essential for developing effective adaptation strategies.

This study presents the implementation and application of an integrated climate-phenology adaptation support modelling system. This is based on regional CORDEX climate models and the CERES Maize model from the DSSAT platform. Novel modules for optimal management and genotype identification under climate change have been developed in the system, employing a hybrid approach that combines deterministic modelling with machine learning (ML) techniques and genetic algorithms. This system was run as a regional pilot over Southern Romania, operating in real-time in interaction with users, performing agro-climate projections (combination of fertilization, sowing date, genotype) and providing best crop management simulated under climate change projections. Multi-model ensemble simulations were conducted for two radiative forcing scenarios RCP4.5 and RCP8.5 and twelve management scenarios, yielding novel results for the region. Results indicate a projected decrease in maize yields for the current genotype across all tested scenarios, primarily attributed to a shortened grain-filling period and reduced fertilization efficiency under warmer conditions. The analysis warns about a projected narrowing of the agro-management options for maintaining a high yield level. However, we find an added value from the impact of genotype selection in mitigating climate change impacts, even in extreme years. Genotype optimisation across six crossed cultivar dependent parameters revealed that while maximum yields decline, specific genotype windows exhibit increased intermediate yields under future climates compared to current conditions. Sensitivity analysis identified the thermal time requirements during juvenile and maturity stages as the most critical factors influencing genotype performance under warmer climates.

This research demonstrates the added value of combining deterministic and data-driven modelling approaches within a coupled climate-crop system for developing effective adaptation strategies, including optimised fertilization pathways that contribute to climate change mitigation.

## 1. Introduction

According to the IPCC (2022), climate change is unequivocal, and its impacts appear more worrying and complex today than decades ago. While research on the effects of climate change on crop yields and agricultural harvests has advanced (Arnell and Freeman, 2021; Hatfield et al., 2020; Rezaei et al., 2023), translating these findings into actionable solutions and scales remains a challenge. This is primarily due to the high complexity of factors that intervene at the local scale of the crop (Eyring et al., 2021; Malhi et al., 2021) including sensitivities of the exchanges to variations in climate sub-components as atmosphere / soil/ biosphere's ecosystems under climate change, natural causes and human activities (Wheeler and Braun, 2013; Xie et al, 2023).

Given the projected global population increase estimated in scientific reports to over 9 billion by 2050 (Godfray et al., 2010), global food production would have to increase by 70-100% to meet the growing demand (Smil, 2005; Selvaraju et al., 2011; World Development Report, 2008). This challenge is further compounded by the agro-climatic conditions expected to become vulnerable and gradually decline due to climate change, particularly impacting water availability (Stehr and von Storch, 2009; Van Ittersum et al., 2013; Villalobos et al., 2012). Another challenge of the problem comes from the need that approaches, and sustainable solutions must not only address the needs of agricultural producers but also align with climate change mitigation goals for 2050, aiming for climate neutrality (Mitchell et al., 2021; Semenov and Stratonovitch, 2015).

Early studies investigating the impact of climate change on crop yields emphasized the necessity of high-resolution modelling approaches. These models should accurately represent management practices and the local effects of climate variables, such as temperature and precipitation (Adams et al., 1998; McKee et al., 1993; Trnka et al., 2014). These affect thermal and water stress and plant physiological processes like stem water potential, stomatal opening, leaf transpiration efficiency (Espadafor et al., 2017). At the regional scale, the relationship between crop yield and water and thermal availability may exhibit strong dependencies on the crop type, geographical location, temporal scale, plant developmental stage and management (Berti et al., 2019; Ceglar et al., 2020; Marcinkowski and Piniewski, 2018; Webber et al., 2018, 2020; Wu et al., 2021). For instance, simulations conducted by Kothari et al. (2022) in regions with arid climates, indicated for future climate change a significant (~30%) decrease without adaptation, but a potential increase (15%) in corn yields under irrigated or under radiation-based genotype efficient use. These findings underscore the critical need for regional simulations that incorporate phenological characteristics with accurate soil moisture estimates to evaluate the effectiveness of various irrigation strategies under future climate scenarios.

In addition to atmospheric conditions, soil properties significantly influence plant growth. These influences occur through physics-based interactions with climate and through alterations in soil chemical composition. Rising air temperatures have been shown to impact the soil carbon budget, with a decline in soil carbon potentially affecting plant and root processes, biochemical cycles, and species composition (Patra et al., 2022).

Crop modelling at local, regional and global scale has significantly advanced, enhancing our understanding of crop systems and enabling the simulation and projection of future yields. These simulations  (Chen and Tao, 2022;  Schauberger et al.,

2020; Tao et al., 2009; Tsvetsinskaya et al, 2001) consistently project global mean harvest reductions with differences in the regional pattern of climate change impact on crop and harvest (Asseng et al., 2015; Li et al., 2022). Not only projected spatial but also temporal impact of changes appears larger and accelerated, motivating intensified efforts on seasonal and multi-annual predictions of plant development and harvest (Baez-Gonzalez et al., 2005; Dainelli et al, 20222; Jin et al., 2022). Analysis of these simulations emphasized also the need to include crop uncertainty in climate scenarios assessments (Basso et al., 2019; Chapagain et al., 2023; Meehl et al., 2007; Rosenzweig et al. 2013).

Meanwhile, model simulations emerged as useful tool in plant breeding analysis (Banterng et al, 2004; Bernardo, 2020; Cooper and Messina, 2023; Mamassi et al., 2023), supporting the development of superior genotypes and breeding methods for maximizing crop performance. These simulations have proven effective in guiding cultivar selection through techniques such as parental selection and breeding by design (Peleman and Van der Voort, 2003; Qiao et al., 2022).

In most recent years climate-crop modelling extended from deterministic crop models (Boogaard et al. 2013; Morell et al., 2016) to data-driven techniques approaches for assessing crop response to weather and climate change (Chang et al., 2023; Meroni et al., 2021; Morales and Villalobos, 2023; Schwalbert et al., 2020; Zhuang et al., 2024). Statistical methods as well as machine learning (ML) used for crop forecast and modelling were however shown to bring for now, limited benefits (Paudel et al., 2023), pointing to possibly hybrid techniques that include physical process in the modelling as a key approach for this challenging issue. On the other hand, breeding optimization techniques using fully deterministic model simulations require a huge number of simulations, analysis and inter-comparisons of predicted crop performance (Pfeiffer and McClafferty, 2007; Wang et al., 2023).

Here we present a novel hybrid approach developed in the frame of the PREPCLIM ("Preparing for climate change") project in which we solve plant phenology using deterministic modelling and merge this technique with an on-line ML-genetic algorithms (GA) iteratively selecting along simulations the multiple parameter range of crop cultivar parameters, according to user-defined criteria for optimal target. The GA simulates the evolution of a population by applying in iterations, genetic operators (selection, crossover, mutation) to a set of candidate solutions (chromosomes). The chromosomes represent potential solutions to the problem and are encoded as strings of binary or symbolic values, with their fitness assessed by a problem-specific evaluation function, here user-required based. GAs have demonstrated success for optimizing agricultural practices using models like DSSAT for irrigation and fertilizer applications (Bai et al., 2022; Wang et al., 2023).

The hybrid approach implemented in this work, focused on ideotype identification, presents the advantage of physically treating the crop complex process along optimizing iterations, thus allowing specific inclusion and understanding of physical causes of responses and of the optimal paths in various climate and management scenarios. Furthermore, it enhances the ability of choosing optimum conditions from continuous multi-dimensional intervals for gene parameters, as opposed to discrete sets. The continuum values approach is an important feature mainly for isolated extreme yield detection, or broader parameters' range and high non-linearity, both aspects of increasing relevance in the context of climate change. Our findings suggest a narrowing of agro-management adaptation opportunities under warmer climates, further emphasizing the importance of this hybrid genotype-agro-management approach to support finding solutions for the future.

The developed system aims to provide efficient and operational support for farmers and stakeholders. It leverages the state-of-the art DSSAT model, a widely used and extensively validated platform for agricultural modelling across diverse applications. The DSSAT model, incorporating complex parameterizations for soil processes, surface-atmosphere exchange, plant development stages, and their interactions with climate and management practices, undergoes continuous refinement through ongoing research and regional calibrations. For this study, the model was specifically adapted to the unique soil characteristics of the pilot region, including parameters such as porosity, composition per soil layers, and thermal properties. Section 2 presents the developed system and its data flow. Section 3a presents results obtained using the system to simulate projected changes in plant phenology and crop parameters for the target region, under various climate and management scenarios, for the current control genotype. Section 3b discusses results obtained using the system's genotype optimization package along agro-management scenarios. Finally, Section 4 presents perspectives and conclusions.

## 2. Data and methods

### 2.1 Study region

Recent observations indicate the Southern Romania as being one of the main hot-spots of climate warming in Europe in summer, with high and persistent extreme heat stress and drought being observed (Copernicus report, 2024). Further, projections of climate for the region show an amplification of this response in climate scenarios, mainly in RCP8.5 (Fig. S1a). For this region, also total precipitation is projected to decrease, while there is an enhancement of extreme precipitation occurrence and a time shift towards late spring (Fig. S1b). These tendencies are increasingly threatening agro-climate conditions in the region, projecting a warmer and drier climate with enhancing extremes.

### 2.2 Scientific approach

Projected changes in agro-climatic parameters for Romania were assessed under two Representative Concentration Pathways (RCPs): RCP4.5 and RCP8.5. These changes were computed as anomalies relative to historical simulations (Hist) using an ensemble of three CMIP5-CORDEX (Benestad et al., 2021; Taylor et al., 2012) high resolution (11 km) climate models, based on the CNRM, EC-EARTH, and MPI global models coupled to the regional climate model RCA4. Subsequently, the DSSAT crop model (Hoogenboom et al., 2019; Jones et al., 2003) was employed to simulate projected changes in phenological and harvest parameters. The DSSAT model was driven by atmospheric conditions derived from each model of the ensemble for the historical period and for the two RCP emission scenarios.

A software package was developed for the DSSAT model that performs identification of optimal model parameters based on user-defined: criteria for optimum, climate-management scenario, region, and time horizon. Optimization goals include maximizing harvest, ensuring stable yields over time, and minimizing nitrogen leaching beyond the root zone (reducing

water pollution risk). Management scenarios allow users to explore optimal cross-combinations of sowing dates, fertilization amounts, and genotypes.

Five main cultivar-specific parameters (P1 to P5) characterizing the maize genotype were analysed across wide ranges of physically realistic values, considering both current and extreme future climate projections for the target area. P1 represents the thermal time from seedling emergence to the end of the juvenile phase, ranging in these simulations from 100 to 500-degree days above a base temperature of 8°C. It significantly influences crop flowering times (Liu et al., 2020), water availability, and ultimately, yield. Studies have shown that utilizing longer-season maize cultivars (dependent also on P1)
can lead to increased harvest in humid regions but decreased harvest in semi-humid regions (Mi et al., 2021).

Longer days increase the period of plant development only up to a threshold value, here 12.5 hours. When the light period of 24h cycle exceeds the threshold of 12.5 hours the advancement towards flowering may be delayed in a measure that it is genetically controlled. P2 measures (in days) the delay in plant growth for each hour of photoperiod above this threshold, and here is ranging in simulations from 0.1 to 2.6 days. P2 influences the flowering time (Langworthy et al., 2018) and the
145 rate of plant development, with long-day plants exhibiting faster development under longer day lengths (Angus et al., 1981). Some tropical maize cultivar need longer nights to flower (short day plants). Related to these, studies have demonstrated the significant role of P2 in mitigating the negative impacts of waterlogging in warmer climates (Liu et al., 2023). P3, the thermal time from silking to physiological maturity, here tested for values from 500 to 1500-degree days above a base temperature of 8°C, significantly influences maturity dates. It also has a main role in plant stress levels (longer-maturity
hybrids increase harvest but under water stress it may provide lower yield (Grewer et al, 2024; Su et al., 2021)) and grain moisture at maturity (Tsimba et al., 2013). P4, the kernel filling rate parameter (ranging from 6 to 12 mg/day), influences grain filling duration, desiccation, moisture at maturity and harvest (Chazarreta et al., 2021). P5, the phyllochron interval, the thermal time between successive leaves tip appearances (expressed in degree-days above a base temperature of 8°C, ranging in these simulations from 3 to 70°C- days), is a critical parameter for estimating the duration of vegetative development
(Birch et al., 1998; Xu et al., 2023). P4 and P5 are important parameters of optimal pant adaptation to climate conditions, since they are drivers of the phenological response and yield formation, in conjunction with the temperature, radiation, humidity, water stress. These genotype (or cultivar specific) parameters are the primary ones considered in DSSAT model parameterizations for plant development processes (Hoogenboom et al., 2019).

The parameter ranges were selected based on extensive genetic database of the original model, and here extended in order to
160 allow investigation the extreme changes induced by climate scenarios. The control values for these cultivar-specific parameters belong to hybrid PIO 3475: P1=200, P2=0.7, P3=800, P4=8.60, and P5=38.90. All the simulations for combinations of parameters values (cross-parameter simulations) were performed under Hist, RCP4.5, and RCP8.5 emission scenarios. For each scenario, crop projections simulations were conducted for twelve agro-management scenarios (Table 1) consisting of sowing date changes and fertilization treatments, at values characteristic for the region after the year 2000
(Table 1a), for each model of the ensemble. Then, for the genotype sensitivity simulations (e.g. the optimal crop response to genotype) we have chosen a lower fertilisation (Table 1b), already used in the region before the year 2000 (when the number

of subsistence farms was high), value aimed for potential mitigation (Ursu, 2025), in synergy with genotype selection. It was shown that Romania, with a fertiliser consumption of 46 kg/ha, had an efficiency comparable to countries with much higher consumption, indicating a significant regional potential for improvement without increasing environmental pressure (Ursu, 2025).

The twelve agro-management scenarios encompass four sowing dates (spaced five days apart) and three fertilization levels (zero, the regional reference value and its double, Table 1). For each agro-management scenario, genotype optimization (finding the optimal set of parameters values under given climate -agro-management and optimum criteria) was performed using two methods: 1) discretized parameter-space runs with subsequent post-processing ordering, and 2) continuum parameter-space search with iterative selection during simulations, employing genetic algorithms (GA). The optimization can be performed for each year, allowing the optimal management and cultivars to evolve over time, and also allowing further investigations of response e.g. during critical years, or in clusters of climate conditions or time-slices, or ensemble means.

The GA-based method employs an iterative approach. It commences with an initial population of randomly generated solutions (chromosomes) and undergoes iterative cycles (generations). In each generation, a selection process is performed to choose the fittest chromosomes for reproduction, based on their fitness scores. Subsequently, crossover (recombination) and mutation operators are applied to the selected chromosomes, generating offspring that inherit traits from their parents. The new offspring replace some of the least fit individuals in the population, ensuring that the average fitness of the population improves over time. The convergence of the GA toward an optimal or near-optimal solution is achieved by balancing exploration (searching the problem's space for diverse solutions exploiting promising regions) and exploitation (refining the best solutions found so far).

2.3 The Software

Here GA has been newly applied to develop an innovative crop selection algorithm, optimizing genotypes across various agro-management scenarios. Steps along the workflow of ML algorithms for optimal genotype identification are:

1. Start with 10 randomly chosen solutions within the bounds of P1-P5;

2. Calculate the mean and standard deviation of harvest of each solution for the time slice;

3. Calculate fitness = (mean of harvest) – (standard deviation of harvest)/4;

4. Randomly choose 4 pairs of 'parents', with the probability being chosen weighted by the fitness;

5. For each pair of parents A and B, create identical children 'a' and 'b' to the parents, then choose a random number of P's to be subjected to crossover, called x;

6. For each child, modify Px as follows:

$Pxa = \text{round} (B * Pxa + (1 - B) * Pxb )$;

$Pxb = \text{round} (1 - B) * Pxa + B * Pxb )$

7. Where Pxa is the value of the x parameter of child "a" (initially identical to that of parent A), and B is the blending factor, set in this paper to 0.75. This technique is called blending, and it generates offspring chromosomes that inherit real-valued traits from both parents while exploring the search space between the parents' positions. The blending crossover promotes a smoother and more gradual search for optimal solutions in continuous domains.

8. Then take each child, and with a probability of 0.5 perform a mutation on one of its chromosomes. This means setting one of the P's to a random value between its allowed minimum and maximum.

9. At this point the children have been fully constructed. Discard the 8 parents with the lowest fitness and substitute them with the children.

10. Repeat from 2.

The system generates output data (agro-climate and optimal paths of cultivars and agro-management) which is disseminated on two platforms (Fig. 1). One is a platform (Info-Platform, Fig. 1a) providing agro-climate information at local scale (NUTS3 level, aligned with the European Union's Nomenclature of Territorial Units for Statistics, primarily corresponding to county level in Romania) over the region. It delivers pre-computed regional climate and agro-climate indicators (e.g. drought, soil moisture, evapotranspiration, aridity indices, storminess, model-based phenological dates, yield), indices of agro-climate extremes (e.g. extreme precipitation frequency and intensity, extreme temperature, scorching index, wind gust, number of freezing / icing days, diurnal temperature range, biological effective degree days) based on observations, re-analysis and climate scenarios for future projections for the region. This platform is publicly accessible <https://climatologis.shinyapps.io/PrepClim/>.

The second platform (User-Platform, Fig. 1b) is an operational, online, user-interactive (two-way) in real-time component, where user requests are submitted, processed as input to the modelling chain and results delivered back to the user for a new, refined request. The access to this user-platform, hosted on an internal server is granted at request.

The core of the modelling system integrates the DSSAT crop model (running on Linux OS) with regional climate models (Fig. 2), with a pre-processing pack developed for coupling. This coupled system incorporates new features, that include the ability of conducting parameter-varying cross-simulations and advanced algorithms for identifying optimal agro-management practices and genotype selections along simulations.

The DSSAT code used in PREPCLIM project, the PREPCLIM software and a PREPCLIM sample data set are available on ZENODO (DOI 10.5281/zenodo.13145521, DOI 10.5281/zenodo.13132587 and respective DOI 10.5281/zenodo.13133107)

a)

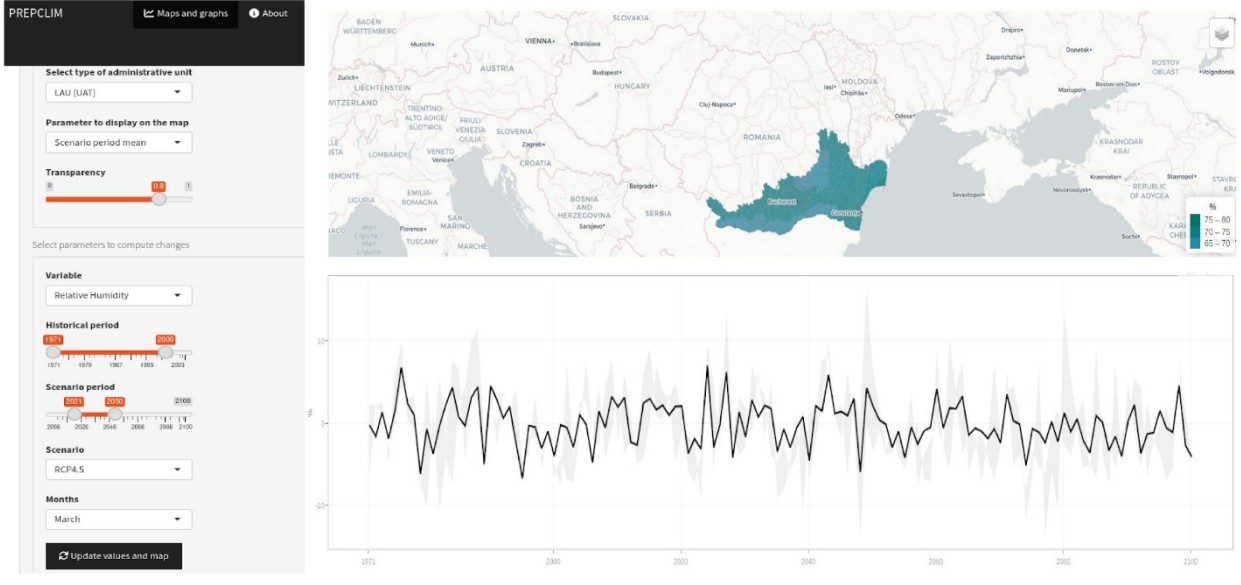

b

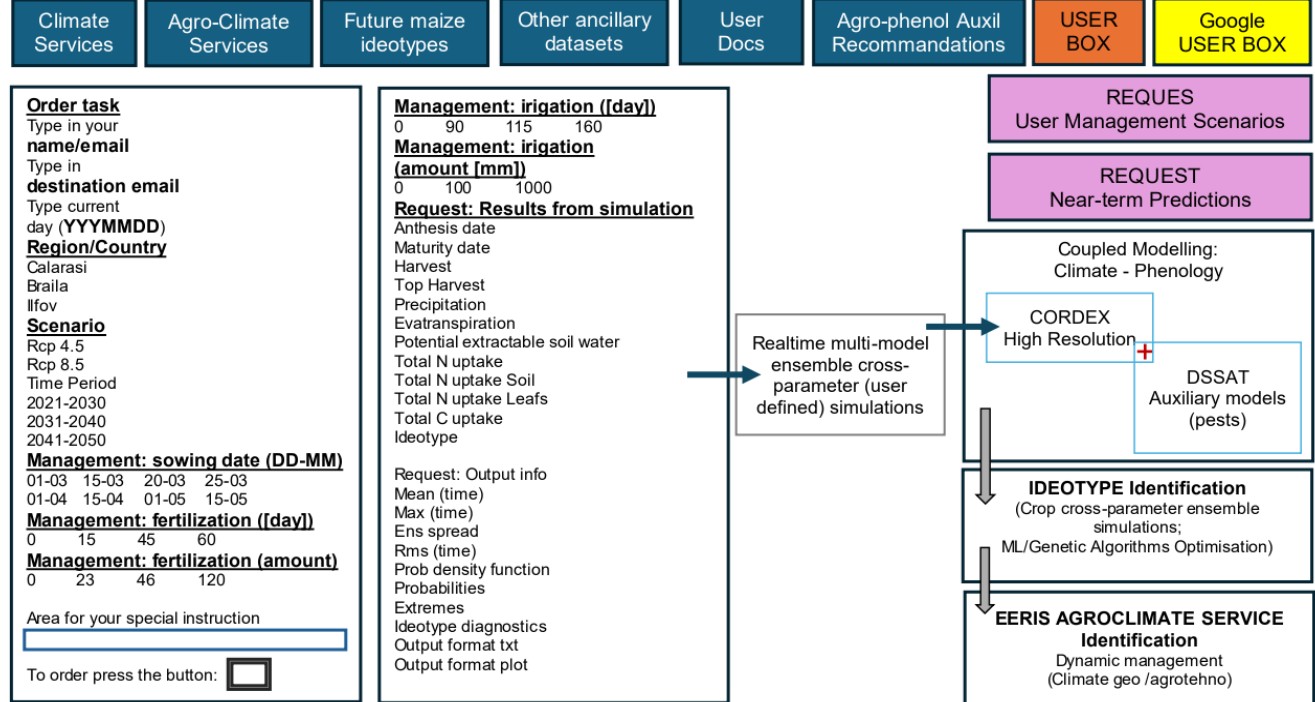

**Fig. 1** a): Info-Platform: Provides local-regional scale information derived from high-resolution regional climate models (CORDEX), e.g. climate, agro-climate data and indicators, indices of agro-climate extremes at the NUTS3 level. b): User-Platform for adaptation support: Processes in real time specific user requests, and simulates management scenarios, identifying optimal paths: Users input parameters (left, e.g: region, period (present / future climate scenarios), management options (e.g. sowing date, fertilization/irrigation time and amount,

genotype); System Output (right, e.g: harvest, projected phenology dates, precipitation/evapotranspiration, Nitrogen and Carbon balances, optimal management paths (dates and management actions), optimal genotype) estimated from ensemble simulations.

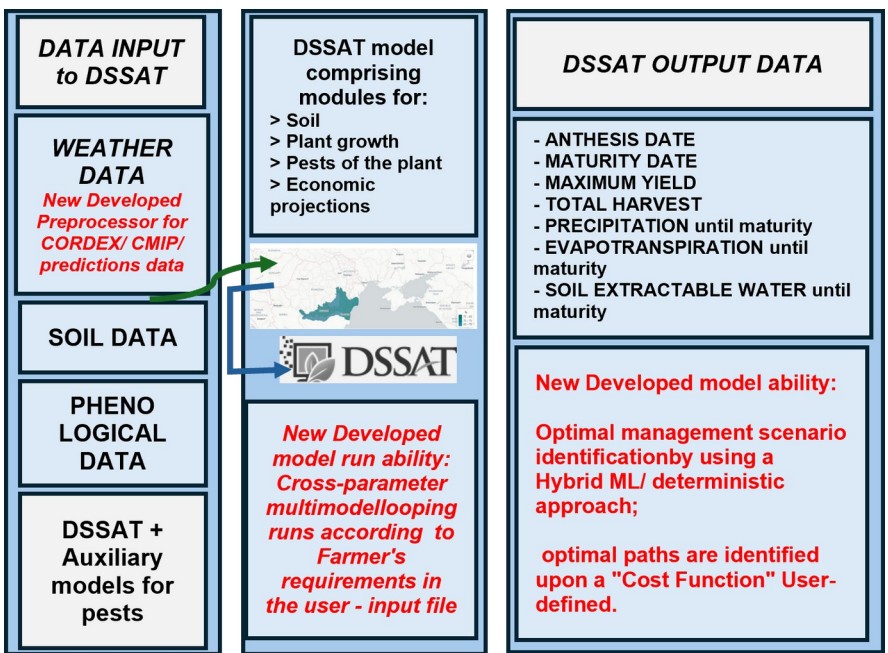

**Fig. 2** The PREPCLIM-v1 work schema: DSSAT-core and modelling components (middle), and data flow: input data (left), output information (right). Red modules were developed in PREPCLIM-v1.

The system was implemented and validated over Southern Romania, target agricultural area, for maize. Potential beneficiaries include researchers, farmers, policymakers, and maize breeders. The system can also assist maize breeders in adapting to climate change by enabling them to evaluate and select genotypes more resistant to challenging climatic conditions. Given the accelerating pace of climate change, such a system may provide valuable support in numerous ways.

**Table 1.** The agro-management treatments: each treatment is described in terms of the sowing date and Nitrogen fertilization amount. In function of the experiment type: a) the experimental set-up for crop phenological projections has: Fx0=0 (no fertilisation), Fx1 and Fx2 (the double of Fx1, 120kg/ha), for each treatment (TR); b) the experimental set-up for genotype optimisations has: Fx0=0, Fx1=23; Fx2=46 kg/ha, values used before 2000, for each treatment (GTR). Sowing date format is "DD.MM"

a) The experimental set-up for crop phenology projections

| Treatment | TR1 | TR2 | TR3 | TR4 | TR5 | TR6 | TR7 | TR8 | TR9 | TR10 | TR11 | TR12 |
|---|---|---|---|---|---|---|---|---|---|---|---|---|
| Sowing date | 1.04 | 15.04 | 1.05 | 15.05 | 1.04 | 15.04 | 1.05 | 15.05 | 1.04 | 15.04 | 1.05 | 15.05 |
| Fertilization (experiment) | Fx0 =0 | Fx0 | Fx0 | Fx0 | Fx1= 60 | Fx1 =60 | Fx1= 60 | Fx1= 60 | Fx2 =120 | Fx2 =120 | Fx2 =120 | Fx2 =120 |

b) The experimental set-up for genotype optimisation

| Treatment | GTR1 | GTR2 | GTR3 | GTR4 | GTR5 | GTR6 | GTR7 | GTR8 | GTR9 | GTR10 | GTR11 | GTR12 |
|---|---|---|---|---|---|---|---|---|---|---|---|---|
| Sowing date | 1.04 | 15.04 | 1.05 | 15.05 | 1.04 | 15.04 | 1.05 | 15.05 | 1.04 | 15.04 | 1.05 | 15.05 |
| Fertilization (experiment) | GFx0 =0 | GFx0 | GFx 0 | GFx0 | GFx1 =23 | GFx1 =23 | GFx 1=23 | GFx1 =23 | GFx2 =46 | GFx2 =46 | GFx2 =46 | GFx2 =46 |

## 3. Results

### 3.1 Model validation

Model validation was conducted using Control simulations (Ctrl) driven by ERA5 reanalysis data (Bell et al., 2021) for each treatment outlined in Table 1a. These simulations, spanning the period 1976-2005, demonstrate the model's ability to capture inter-annual variability in harvest yields, including both high and low yield years, when compared to the measured available data for the region (Fig. 3). The amount is more challenging for validation due to time-evolving constraints over the region. Some contributions were identified, as large variations in fertilization over 1990-2000 with an abrupt decay after 1991, then followed by an increase around 2000 (Popescu et al, 2021), variations in the available field machinery, pest and weeding, and lack of counteracting methodology (Fig. S2). However, these are traceable in these simulations' comparisons (that show lower skill about 1995, for which it was reported a minimum of fertilizer plant protection equipment (National Institute of Statistics, 2025, http://statistici.insse.ro)).

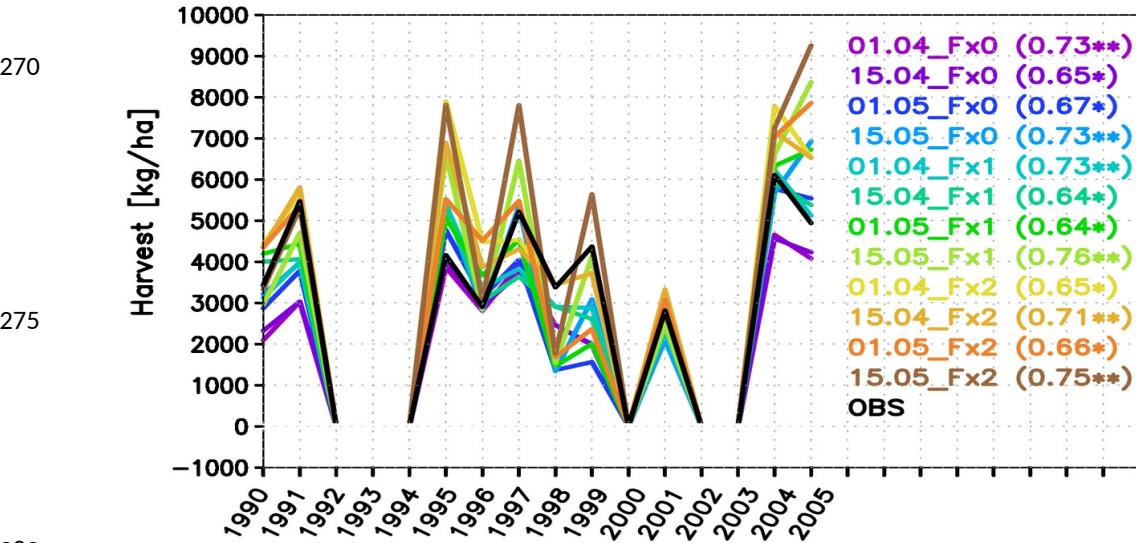

**Fig. 3** Simulated (colours) vs. measured (black) harvest in southern Romania for 12 management scenarios (Table 1a). Right: treatment defined by sowing date and fertilisation (Table 1a) and Pearson correlation between simulated treatments and measured Harvest (** p<0.05, * p<0.10; zero are missing values).

## 3.2 Phenology and Harvest Projections for the Control Genotype

Projected changes in phenology for the control genotype (Pioneer 3475) were simulated using the DSSAT model under historical (Hist) and multi-model climate projections of RCP4.5 and RCP8.5 scenarios. Further, multi-genotype simulations are discussed in Section 3.3.

### 3.2.1 Phenology dates - projected changes

Ensemble model simulations provide projected changes in phenology, for the control genotype, under different fertilization levels (0, 60, 120 kg/ha, Table 1a) and sowing dates, averaged over 30-years, in scenarios (2021-2050, RCP4.5 and RCP8.5), versus Hist. Figure 4a, b illustrates the ensemble model changes, demonstrating an earlier anthesis date by up to ~6 days and an earlier maturity date by up to ~10 days across all scenarios. These time-shifts result in a shortening of the grain-filling period by up to 10% across the ensemble, and are a consistent response observed in each individual model. Early sowing dates exhibit a more pronounced earlier shift in anthesis under warming scenarios, a response even more pronounced under RCP8.5.

a)                              b)

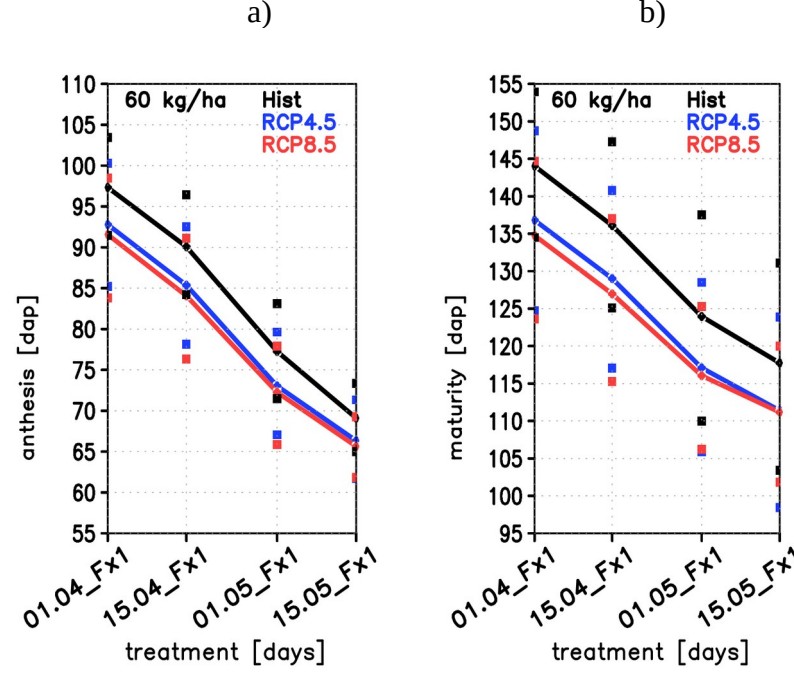

**Fig. 4** Simulated a): anthesis dates ([dap], days after sowing) and b): maturity dates ([dap]), under historical conditions (black), RCP4.5 (blue), and RCP8.5 (red) scenarios. Results are shown for the four sowing dates and nitrogen fertilization level of 60 kg/ha (Table 1). The maximum and minimum value over the ensemble members for each treatment and climate is shown (dots) and the lines represent the ensemble mean for each treatment and climate simulation.

Under warmer climates we note more frequent occurrences of critical situations with suboptimal grain filling and potential crop failure, under fertilization. These were linked in previous studies to non-linear interactions between fertilization and temperature (Huang et al., 2024) with excessive fertilization during reproductive stages under elevated temperatures potentially inducing higher stress conditions.

In our study premature ending of simulated vegetation season occurred more frequently in treatments with higher nitrogen fertilization. This may favour leaves development, enhanced transpiration and earlier depletion of the soil moisture leading later to water stress. However, this lead in average to small mean changes in maturity days and in grain filling duration (Fig. S3).

### 3.2.2 Harvest - projected changes

For harvest, the ensemble simulations project an overall decrease under both RCP4.5 and RCP8.5 scenarios and across all sowing dates and fertilization levels (Fig. 5), compared to the historical period.

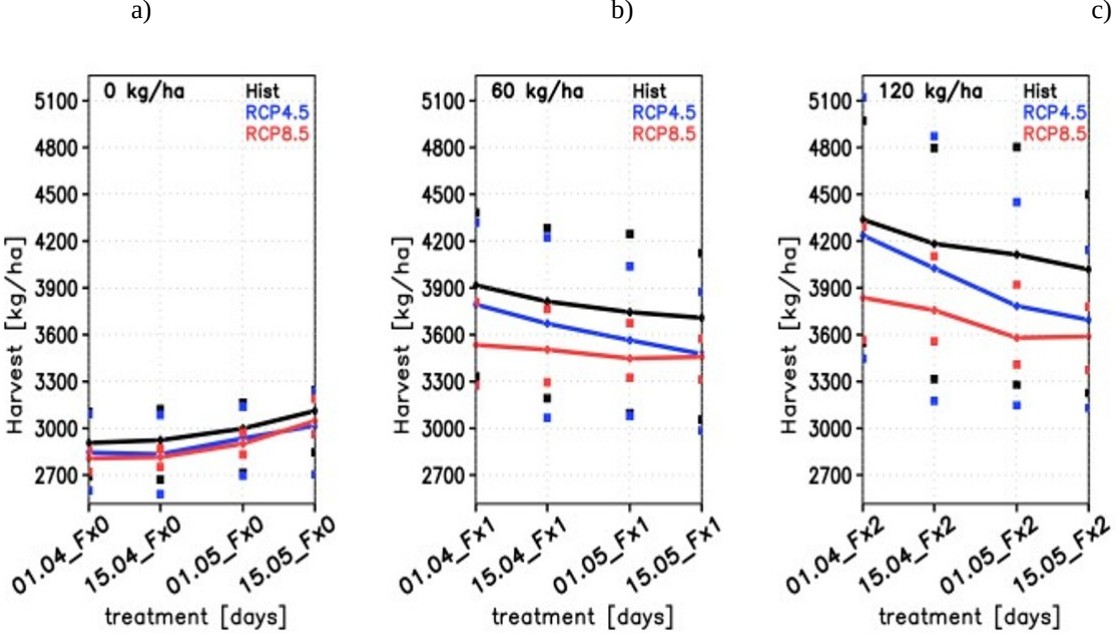

**Fig. 5** Simulated Harvest (kg/ha) under Hist (black), RCP4.5 (blue) and RCP8.5 (red) scenarios, for four sowing dates across three fertilization levels (Table 1a): 0 (a), 60 (b), and 120 (c) kg N/ha (from left to right). The maximum and minimum value over the ensemble members for each treatment and climate is shown (dots) and the lines represent the ensemble means.

Harvest decline in climate scenarios is related to several factors: 1) reduced rainfall during the growing season (Fig. 6), as evidenced by a strong correlation (0.5 in April to 0.8-0.9 in July-August, over 30 years, Fig. S4) found between harvest (H)

and accumulated precipitation in the Ctrl and in future simulations; 2) a shortened grain-filling period due to a projected earlier flowering and an even earlier maturity across all the models (Fig. 4), potentially limiting biomass accumulation; and 3) decreased fertilization efficiency under warming conditions, in the sense that the difference of harvest in Hist versus in scenario, increases (non-linearly) with enhanced fertilisation (Fig. 5). Hence, the same increase in fertilisation may bring less benefit in a warmer climate. This benefit for H is of about 10% in Hist versus 7.6% in RCP8.5 for early sowing and about

8% in Hist versus 4.3% in RCP8.5 for later sowing for doubling the N amount of nitrogen (Fig. 5 b, c). This efficiency decay feature underscores the primacy of reduced accumulated precipitation (Fig. 6) and of higher temperature, that lead to a non-linear H response to fertilization (Huang et al, 2024). This has an impact on Harvest maximal range and further analysis on the change in intermediate and extreme harvest values can be found in Sect. 3.3.1. Their influence is noticed as well in the absence of fertilization (Fig. 5a), when H still declines in warmer climates, with a dominant control from precipitation. The

correlation along sowing dates between H and accumulated precipitation until maturity (Pmat, Fig. 6), is r(H, Pmat) >0.96 in both scenarios.

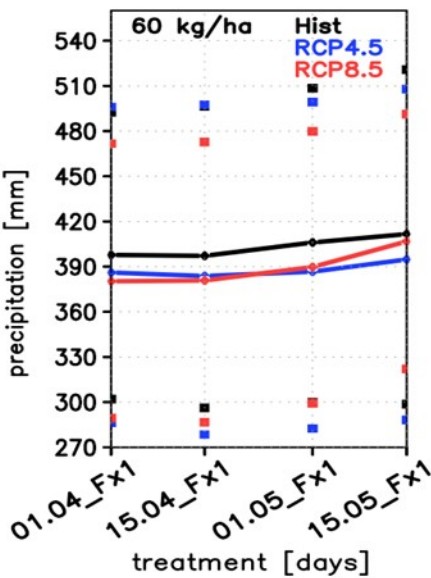

**Fig. 6** Precipitation accumulated until maturity (mm) (legend as in Fig. 5)

The role of the precipitation timing is emphasised: for late sowing, RCP8.5 shows more accumulated Pmat (and H) even in cases of a shorter accumulation season (Fig. 4) but having projected a precipitation increase towards late spring (Fig. S1b), that may significantly favour critical growth stages.

### 3.3 Optimal Genotype Identification

The system was further developed to extend the management scenarios for multi-genotype simulations and implement methods to identify ideotypes under each agro-climate scenario. The aim is to search for management scenarios that yield optimal outcomes defined by user-criteria such as maximizing harvest yield, stabilizing yield, or minimizing pollutant emissions. Two optimization methods are implemented: a discrete-parameter, purely deterministic technique, and a hybrid approach that combines deterministic modelling with continuous-parameter Machine Learning-based Genetic Algorithms for iterative genotype selection.

The deterministic method involves conducting multiple-genotype crop model simulations, with optimization performed as a post-processing step. Genotype parameters are defined within pre-established limits and discretization. Multi-model simulations are then performed, where each parameter is individually varied while the remaining parameters are held constant. The total number of simulations in this case is determined by the chosen discretization level. In contrast, in the hybrid technique the parameters values are selected from a continuous range of values, identifying and iteratively improving the best sub-domains.

#### 3.3.1 Optimal genotype under climate change

**i) harvest as a function of the genotype H(G) in scenarios versus current climate**
We analyse the distribution of H obtained along multi-genotype simulations, ordered from maximum to minimum values and denote the genotypes corresponding to this ordering "H-ordered genotypes", which is simulation (model, scenario) dependent. Comparing these H distributions for the two climate scenarios against Hist, indicates projected changes in the ensemble-model PDF (probability density function) of H under warmer climate.

A first outcome demonstrates in Fig. 7a, b that for the H-ordered genotypes, a projected average decrease in Harvest (H) occurs within the range of maximum H values (genotypes in the top H-percentile interval (0%, 2.5%) of the H-ordered genotypes), under both scenarios, and mostly affecting the earlier sowing dates (Fig. 7b). Across models of the ensemble, we note a strong modulation of this behaviour by precipitation (Fig. S4), particularly for unfertilized scenarios. Precipitation exhibits high inter-model variability and significant regional-scale uncertainty, pointing to the need of ensemble modelling for reducing it. Linked to this precipitation response some individual models, in opposite to the ensemble mean, may exhibit even increases, for genotype intervals in the top percentile in Harvest, under climate scenarios (Fig. S5). In contrast, the warming trend is a consistent feature across models in the region, contributing other model-systematic responses such as earlier anthesis and maturity dates and shortening of the grain filling season.

The second note regards a different response projected in the intermediate H values (Fig. 7a, c). Genotypes corresponding to the intermediate H values (genotypes of intermediate H-percentile interval (25%, 75%) of the H-ordered genotypes) show projected higher intermediate H values in climate scenarios than in Hist (Fig. 7c), affecting less the earlier sowing (Fig. 7c).

These together lead to a narrowing of the H-values range of responses, in the top and intermediate H-percentile intervals, to
the same managements applied, in scenarios compared to Hist. Same management spread would lead to closer H-responses,
with enhancing the probability for occurrence of intermediate values and decreasing the probability for highest H values (a
third feature of projected changes).

Finally, we note that despite this narrowing, earlier sowings appear systematically as better timing options (Fig. 7a),
improving by up to 2% in scenarios (respectively to 4% in Hist) unfertilized case and up to 8% in fertilised case in ensemble
time-mean scenarios (respectively to 12% in Hist) (Fig. 7a), with the lowest percentage for RCP8.5. Earlier sowing was
reported in other recent studies as optimal for spring maize harvest (Djaman et al, 2022).

### ii) options for adaptation and mitigation using genotype analysis

These three features of cross genotype-agro-management impact: - projected lower maxims of H in scenarios (mainly for
early sowing), projected higher intermediate H (mainly mid-late sowing); - a narrowing of the range of H in the top and
intermediate H-percentile intervals with higher/ lower probability of intermediate/ high values occurrence, have practical
adaptation outcomes.

Finding mitigation solutions, while preserving yield, e.g. finding appropriate changes in agro-management practice that
allows a lower, less pollutant fertilization, at a same Harvest percentile, appears indeed to be supported by genotype
selection. Fig. 7 (mitigation window shown for RCP4.5) indicates that for a Harvest given percentile, we get intervals both in
the intermediate and in top percentiles where changing the sowing date for a lower fertilization, brings even improved
solutions. These intervals are defined by intersection points of H-curves defining parameter-zones of both mitigation and
optimisation. Alternatively, for a given H values range and treatment, one may estimate the interval of genotype-parameters
to achieve that range, an information useful to improve local crop usage.

Apart from any comparison with Hist, it is important for long term adaptation, that one may find genetic combinations with
high yield in specific target percentile under a given climate (e.g. first 50 values, as in Fig. 7b).

At yearly level, the interest for some of these genotype parameters combinations may increase, providing that distinct
weather favourable patterns will be identified, once with progress achieved in seasonal and multi-annual weather forecasting
(O'Reilly et al., 2025).

a)

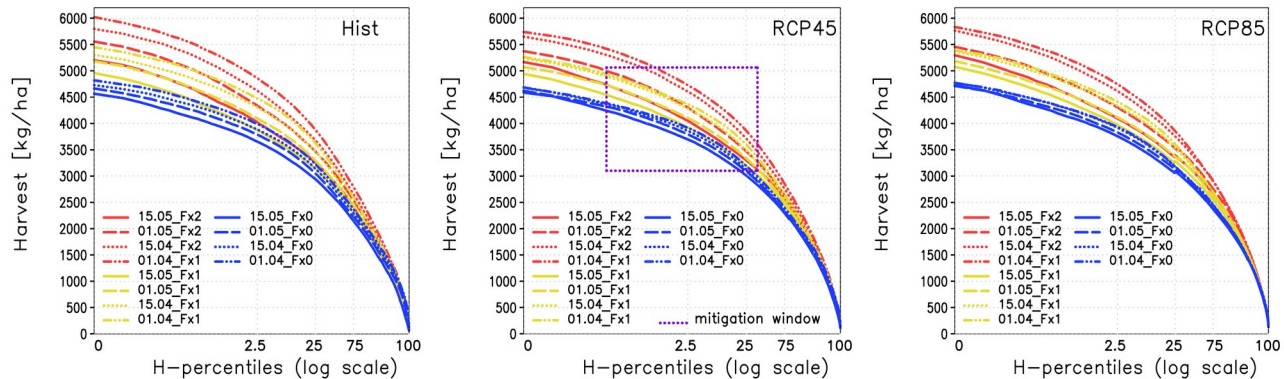

b)

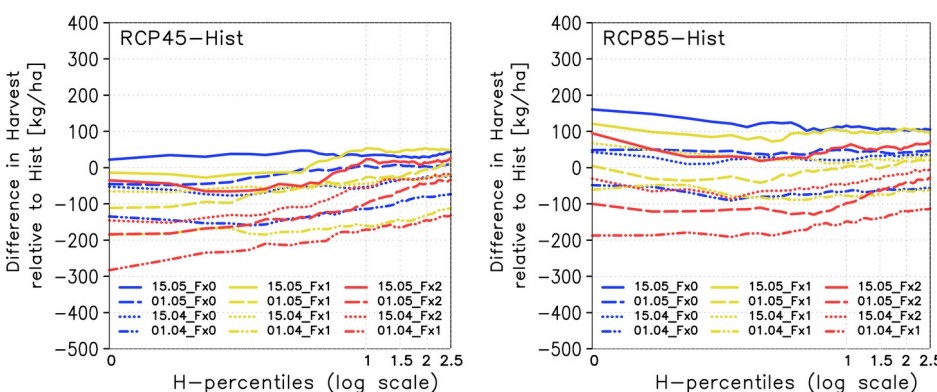

c)

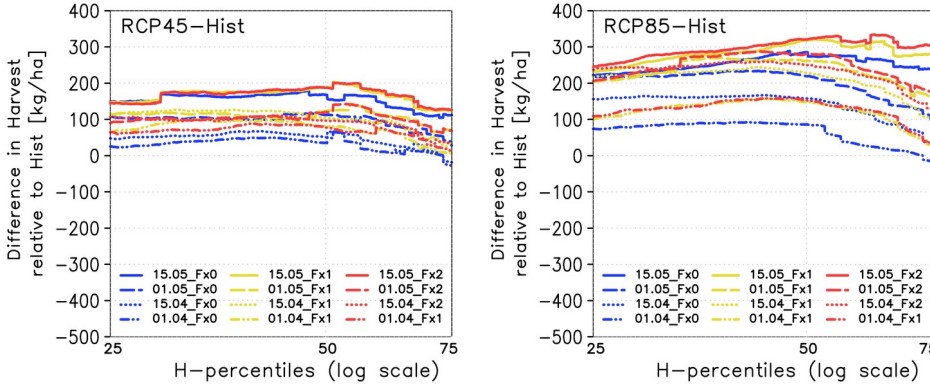

**Fig. 7** a) Harvest multi-model time mean: percentiles of the H distribution ordered from maximum to minimum value (left to right on x-axis, logarithmic scale). The simulations are for: Hist (left), RCP4.5 (middle) and RCP8.5 (right) for multi-parameter genotype changes (for six parameters resulting 1890 simulations), see also the models H distribution in Fig. S5; b) differences in projected harvest for RCP4.5 minus Hist (left) and RCP8.5 minus Hist (right), for the upper H 2.5% percentile (the first 50 values) and for the intermediate H percentile 25%-75% in c) (the 475-1400 H-ordered values). Plum rectangle in Fig. 7a (RCP4.5) shows in simulations, a window of potential actions for mitigation through genotype- agro-management selection (text). Rectangles are defined by the two (extreme) intersections of H-curves for different management scenarios, along x-axis.

### 3.3.2 Optimal Genotype parameters under climate change

#### i) optimal genotype parameters

We further discriminate H response per genotype parameters (P1-P5), to understand the source of the changes in Fig. 7 and the possible adaptation paths under climate and management scenarios.

Parameters' analysis (Fig. 8) shows that in all simulations, higher top harvest is obtained under: shorter thermal time from seedling to juvenile phase (lower P1, Fig. 8a), shorter photoperiod-delay (lower P2, Fig. 8b), slightly shorter thermal time between successive leaves appearance (phyllochron, lower P5, Fig. 8e in the intermediate H-percentile interval but longer in the top H%), also longer thermal time to maturity (higher P3, Fig. 8c) and higher grain filling rate (higher P4, Fig. 8d). These results are in coherence with findings along recent works. Shorter P1 or lowering the seedling-juvenile thermal time for increasing H (Fig. 8a) is in agreement with Mi et al., (2021) for semi-humid areas, (the current class of this region, with semi-arid trends projected, Fig. S1a), and the same for P2, while slower maturity (higher P3) and enhanced filling rate (higher P4) being linked to higher kernel weight and harvest in agreement with recent studies (Grewer et al., 2024).

#### ii) changes in optimal genotype parameters in climate scenarios

Comparing the genotype parameters in climate scenarios against Hist, reveals the new plant strategy put in place in the new climatic conditions, for maximizing the harvest. The ensemble simulations (Fig. 8) shows that highest harvests are reached with genotypes that ensure a longer thermal time from seedling to juvenile phase and longer thermal time to maturity in scenarios compared to Hist. To a smaller extent this is also achieved by a longer photoperiod delay P2, higher grain filling

rate P4 and longer phyllochron interval P5, in scenarios than in Hist, for a same percentile of the Harvest. These show that under warmer climate it is essentially important to avoid too fast growth on vegetative and grain filling stages of the development. Indeed, slower development phases occur in scenario simulations with higher H for increased P1 and P3 and

435 related to these, under longer photoperiod (P2 increase). Other contributions come from ensuring a slower rate of appearance of successive leaves (P4 increase), while a higher grain filling rate (P5 increase) appears to partly compensate mainly for intermediate-H, for the negative effect of higher temperature that decreases the seed-filling duration and seeds number and size and finally the harvest.

In other studies, this compensation was shown to be minor compared to the loss of seed-filling duration in warmer climate

(Singh et al., 2013) that points to P1 and P3 as main drivers for Harvest in climate scenarios. Percentages of the parameters' changes in scenarios versus Hist for a given percentile of harvest (Fig. S6) confirm this main driving.

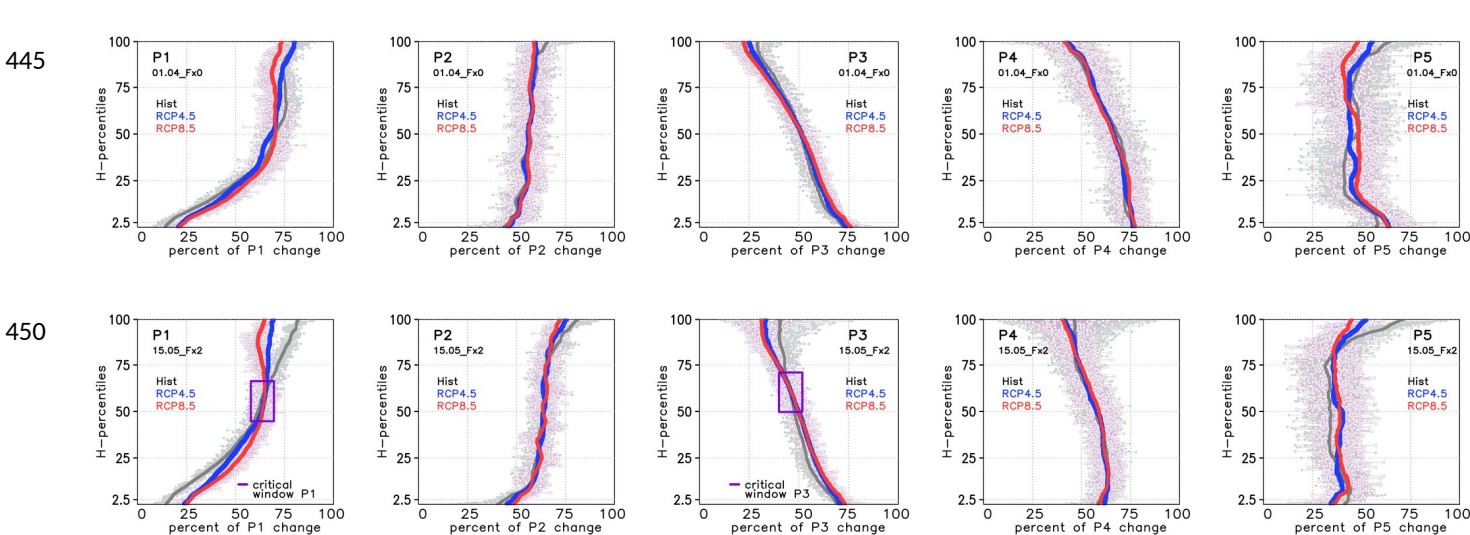

**Fig. 8** Parameters' values corresponding to percentiles of the H distribution (ordered from maximum to minimum H value). On X axis is the percent of parameter change (increase) relative to the maximal change tested for each, normalised to its control value. The figure compares these percentiles for Hist (black), RCP4.5 (blue) and RCP8.5 (red), running mean values (full lines, 100 values window), ensemble mean, time-mean; unsmoothed values are shown by dots only for Hist (grey shade) and RCP8.5 (pink shade), the RCP4.5 values being generally intermediate. Percentiles are from a number of 1890 genotype simulations. These are shown for two treatments

(01.04_Fx0 at top and 15.05_Fx2 at bottom). The plum rectangle indicates a critical parameter range for P1 and P3 (text) defined as +/- 5%H around threshold values (the parameter value at the intersection between scenario and H curves). Thresholds are defined as the (neutral) value of the parameter for which the same H-percentile is reached in scenario and Hist. Thresholds control the limit parameter values for which the scenario leads better/ worse H percentiles than in Hist.

**iii) optimal genotype parameters in management and climate scenarios**

Agro-treatments choice may significantly modulate the H response to genotype parameters. Delaying sowing, requires first gradually decreasing parameters in order to maximize H (Fig. 9, also in Fig. 8), for both Hist and climate scenarios. For P1-P3 this decrease reflects the priority in avoiding a too late end of the juvenile stage (and shift in climate conditions) and a too late (autumn) maturity stage that is slowing the grain filling and leading crop failure.

However, Fig. 9 also shows that these parameter's decreases cease or even reverse under extreme delay of sowing. For highest delays the development stage is getting too short under P1's too strong decrease while daily temperatures becoming higher, hampering the development. The same is seen for the maturity, with P3' too strong decrease favouring a too quick grain filling. Hence the plant strategy for adaptation after a threshold of sowing-delay is similar to the one already seen in its adaptation to warmer climate, in scenarios. Higher harvest is then reached by gradually switching to only moderate decrease or even increases of parameters along with increasing delays in the sowing date.

This gradual change in the monotony for the parameters leading to higher harvest, as a function of sowing delay appears quite systematic for all parameters.

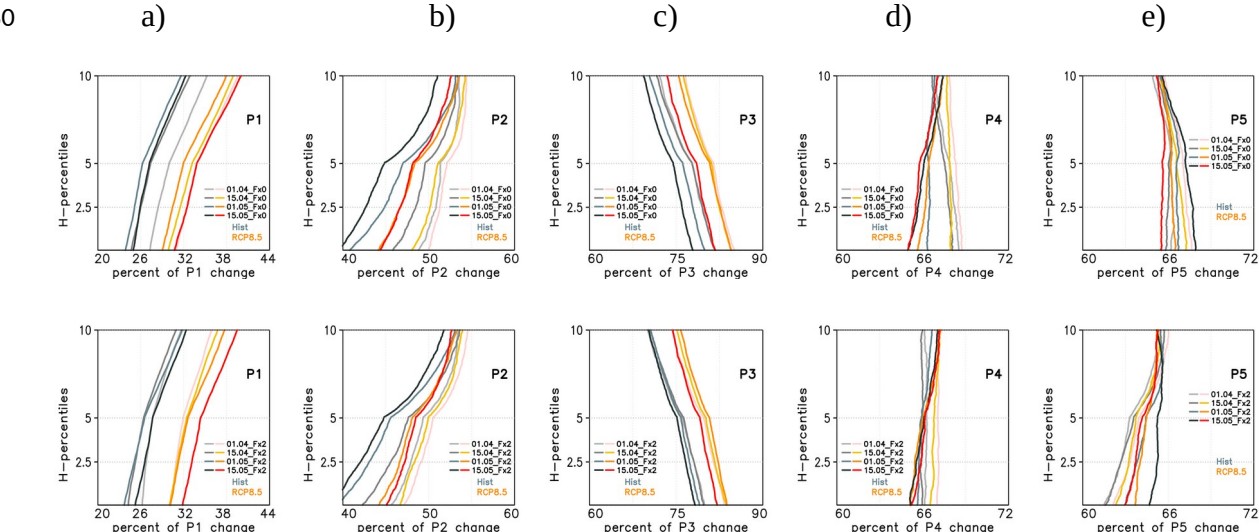

**Fig. 9** As in Fig. 8 but for all sowing dates, no fertilization Fx0 (top) and with fertilization Fx2 (bottom). Parameters are shown for the top 10% highest harvest. On X axis is the percent of parameter change relative to the interval tested for each. Grey colours are for Hist and yellow-red for RCP8.5 (light to dark is from earlier to latest sowing).

This crop adaptation mechanism, converging towards the one projected for climate scenarios, shows that gradually under enhanced warming, the crucial priority in adaptation transfers, from the key issue of ensuring climatological conditions for the development to the key issue of avoiding a too fast growth leading crop failure.

**iv) optimal genotype parameters in adaptation and mitigation strategy**

For each agro-management and climate scenario one can identify threshold values of parameters (defined as the (neutral) value of the parameter for which the same H-percentile is reached in scenario and Hist). Critical windows (shown in Fig. 8 for P1 and P3) are defined at +/-5H% of this threshold. These values depend on the parameter, the sowing date and the fertilization level. When reaching the threshold parameter value, the genotypes lead to the same H-percentile in scenario and Hist. So, the thresholds controls the limit values to which a monothonical change in the parameter in scenario leads better/ worse H percentiles than in Hist. The critical window (Fig. 8, for P1 and P3) indicates the limit and values until which a quantifiable advantage is brought by changes in P1 and P3 in scenarios compared to Hist, information useful for adaptation under climate scenarios.

Second remark is on the probability of an outcome. Since all the slopes of parameters, each as a function of H ordered-values are lower than in Hist (Fig. S7), there is a narrower parameter interval for all those parameters decreasing with H (e.g. P1, Fig. 8a) and a broader one for those parameters increasing with H (e.g. P3, Fig. 8c), in climate scenarios. P3 increases are broadening the interval for H-highest percentile, potentially presenting, in this sense, higher probability than P1, on highest H-values outcome.

The genotyping results were found both in simulations involving deterministic and the hybrid deterministic-ML methods. The hybrid method involved the same cross-simulations, but the selection of parameters values for H optimization and ordering was no more following a pre-defined discretisation but instead a random picking up over a continuous interval of values with successively retrieving the best generation. It applies for optimization, classic Genetic Algorithms methods in which selection of pairs is based on the user-criteria (e.g. maximum harvest, stable harvest, etc.). Our results show that for the same physical intervals of the genotype parameters, the ML hybrid technique only after 20 generations shows at least 50% chances to get a better result than the deterministic model, while after 100 generations, it already increases at 80% chances to get better results with also computational efficiency. CPU time is reduced in this case by more than 30% using the hybrid technique compared to the fully deterministic model on a VM Linux platform. Hybrid method emerges as a better solution since it can identify improved optimums at lower computational prices.

**4. Discussions**

The results found are in line with other results in recent studies, using different approaches and observational data, and offer an extended (continuum-parameter) assessment towards a more generalised frame, allowed by the implemented system. For the plant response under management treatment, delaying sowing limiting elongations of the development phase, was also found in other studies (Huang et al., 2020) to reduce the impact of high temperature increases on Harvest (Fig. 9) and, in some cases, precipitation decrease and water stress. This response was also found stronger under enhanced fertilization and

delayed sowing (Figs. 8, 9). Also fertilization lowering P5 so enhancing leaf appearance rate (Fig. 8e), assessed also in earlier studies mainly for warmer climates (Dos Santos et al., 2022; Sardans et al., 2017) was recently put in relation to P2 decrease (Fig. 8b) mainly along sensitive photoperiods (Hu et al., 2023) and to higher harvest reached, through enhanced evapo-transpiration maximizing the N uptake (Lu et al., 2024). In warmer climate scenarios, limitations in the expansion of new leaves (increase of P5, Fig. 8e, at highest H-percentiles, no-fertilised case) was shown to be an adaptive tolerance mechanism to drought and heat stress conditions (Fahad et al., 2017).

Further, for moderate sowing delay, fertilisation was shown to require slower grain filling (P4, Fig. 9d) under reduced P1, P2 and P3, controlling the N stimulated growth, under hydric stress conditions of current and projected climate for non-irrigated crop (Yang et al., 2024). Under high delay and warmer climate, a higher grain filling is required (Fig. 9d). This increase for P4 under increased warming may reflect an adaptive strategy of plants to accelerate development under drought stress, allowing plants to end their life cycle before impact of severe drought stress occurs (McKay et al., 2003; Roeber at al., 2022).

Simulations here emphasize and compare adaptation paths of gradual plant response to warming climate. These emphasise some reduction in the efficiency of adaptation through crop management in warmer climates. Meanwhile, genotyping shows the possibility of identifying parameters still able to enhance the efficiency of adaptation under climate and agro-management scenarios. The ability of exploring continuum-parameter space not only offers a general picture of adaptation cross-solutions but identifies critical values of the parameters that for small perturbations may lead the system response into different states (thresholds of sowing-delays, or genotype parameter values). Without an integrated modelling approach, estimating or emphasising these points meaningful for adaptation is hard, moreover since these are climate-management scenario dependent.

## 5. Conclusions

The main outcome of this study is that an agroclimatic real-time Interactive Service was implemented towards adaptation support, that allows performing real-time, user-requested, agro-management modelling scenarios for the region, under current and future climate. A novel feature of the system is the ability for identifying optimal management paths answering the user's request, providing optimal cross-cultivar parameters, such as sowing date, genotype parameters, amount and date of fertilization.

The system provides solutions and estimates the associated uncertainty by using multi-model ensembles for each agro-climate and management scenario. The crop optimization criteria are user-defined and can relate to high harvest, stable harvest, low pollution. The optimization module implemented uses a hybrid deterministic - ML methodology. It performs multi-model simulations using physical models of climate and plant penology and optimization is done either through discretizing the parameters' space and optimisation post-processing or using hybrid physical-ML Genetic Algorithms methods. ML methods are spanning continuous parameter's space iteratively selecting along the simulations the best fit

parameters, allowing to identify unprecedented optimal configurations (H maximas), not reachable under the discrete deterministic method. The overall system output information is layered and accessed from two interfaces: one static, for information purpose (phenology, harvest, climate, extremes at high resolution NUTS3 level) and a second is real-time interactive online, through which the user places requests and receives the system-performed management simulations required (including uncertainty along multi-models) and identified optimal paths for adaptation. These platforms are operational for two emission scenarios RCP4.5 and RCP8.5 and twelve management scenarios (sowing dates and fertilization), for the time-horizon up to 2050, with open-source code (EERIS platform). The results of these were discussed in this work for the pilot region South Romania.

For the current genotype, in both emission scenarios it is projected a mean decrease (14% in ensemble mean, with higher values per model) of the projected harvest, for all the management scenarios (sowing-dates and fertilization) tested. This was linked to a projected shortening of the grain filling season (by up to 10% with an earlier shift of both anthesis (5 day) and maturity (10 day) phases), and to a mean decrease of the fertilisation efficiency under climate scenarios, stronger in RCP8.5 emissions.

The impact of genotype perturbations on crop parameters is analysed along six cross-genotype parameter simulations, for the agro-management-climate scenarios. The main questions: i) Can we identify optimal genotype parameters that lead to maximal harvest? How do these differ under projected climate change and/ or under agro-management options and can these enhance our understanding to guide our options? iii) Can be genotyping a (better) solution for adaptation under climate change in the region?

These simulations showed that the maximal H values are projected to decline for all agro-management and breeding simulations performed, in emission scenarios compared to Hist, with a higher decline for earlier sowing. H-values then increase in the intermediate-percentile harvest in scenarios versus Hist and there is enhanced probability in scenarios to reach the historical values in this range through agro-management and breeding. These indicate a narrowing of the responses range to a same agro-management, with lower / higher probability of reaching values in the highest / intermediate H-range in climate scenarios compared to Hist. In practice, these express that we can identify the H-percentile (genotype), where agro-management choices will optimize the outcome compared to Hist, including finding solutions with lower fertilisation, less pollutant.

For effective support in adaptation applications, individual genotype parameters were analysed in climate scenarios versus Hist. This showed that the thermal times to juvenile (P1) and maturity (P3) are key genotype parameters driving harvest changes in the region, requiring increased values in climate scenarios compared to Hist for a same high-harvest percentile range. This range is identified through critical values of the genotype parameters, determined for each treatment and climate scenario. There is significant variability of these cultivar dependent parameters impact under agro-management treatments. Moderate delayed sowing and enhanced fertilisation may diminish the shifts in optimal parameters in scenarios compared to Hist for a same H-percentile, in contrast to extreme managements.

These results show that Genetic approaches offer adaptation strategy support in helping plants to resist drought stress under warming climate, while a projected narrowing of the agro-management options for maintaining a high yield level is emphasised under warmer and drier climate. Moreover, it was shown that the optimization is improved by using a

590 hybrid ML genetic algorithm method coupled to the deterministic model-output, leading to detecting better solutions, under a continuous-parameter space search. The system can be further used for searching paths along extreme drought years, along with irrigation options investigation. Coupled with weather extended predictions (seasonal, year -decadal) this could provide near real-time adaptation support.

**Code and data availability**: The code is available in the Github repository at: https://github.com/pneague/Genetic-Algorithm-for-Corn-Genotype-sowing-Date-Optimization under a BSD 2-Clause Simplified License.

The DSSAT code used in PREPCLIM project, the PREPCLIM software and a PREPCLIM sample data set are available on ZENODO (DOI 10.5281/zenodo.13145521, DOI 10.5281/zenodo.13132587 and respective DOI 10.5281/zenodo.13133107) The Info-platform is publicly accessible <https://climatologis.shinyapps.io/PrepClim/>.

The access to the second platform developed in PREPCLIM (User-Platform, Fig. 1b), hosted on an internal server is granted at request addressed to the correspondent author.

**Author contribution**: MC: model implementation, code for optimal adaptation tool, pre and post-processing, model simulations, results analysis, development of the User-Platform, paper writing; LC: DSSAT model set-up, results analysis,

paper review; PN: ML method implementation and runs, results analysis, paper writing; AD: model validation; VA: development of the Info-Platform; ZC and AI: platforms upload and update; AP: agro-meteorological station data providing; GC: DSSAT model input for the target region.

**Competing interests:** The contact author has declared that none of the authors has any competing interests

**Acknowledgments*:* The authors are grateful to UEFISCDI who provided the financial support of this work under the Project Grant PREPCLIM PN-III-P2-2.1-PED-2019-5302.

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
