# Peer review of "A modelling system for identification of maize ideotypes, optimal sowing dates and nitrogen fertilization under climate change – PREPCLIM-v1"

_Geoscientific Model Development, 2024_

## Referee Comment (RC2)

**Review: A modeling System for Identification of Maize Ideotypes, optimal sowing dates and nitrogen fertilization under climate change - PREPCLIMv1 (gmd-2024-105)**

**General comments**

In this manuscript, the authors describe a model system—including a graphical user interface (GUI)—that can be used to help decisionmakers understand what sort of crop planting practices will be optimal in the future. These practices include sowing date, fertilization level, and genotype selection. The latter is especially interesting, with the authors using a genetic algorithm in addition to a more deterministic method. The system is demonstrated for maize in Romania as a case study, but such tools are of much broader interest.

Unfortunately, the paper is hampered by poor organization, unclear use of language, and low-quality figures. An extensive rewrite is required to address everything from separating methods and results, to improving the way experiments are referenced, to more minor language and typo issues. I thus recommend this paper be reconsidered only after major revisions.

**Major comments**

- What are the genotype parameters that are getting modified? What do they represent in terms of processes?
  - This is only explained deep into the Results section (L516-519). This should be in the Methods instead.
  - o If P4 was kept constant, why is it even mentioned?
- Treatment naming is very confusing, which results in figures that are hard to understand.
  - Looking at Table 1, what is the difference between Fertilization (3N) and Fertilization (1N)? How can, e.g., TR2 get both 60 and 23 kgN/ha? I think, from reading the rest of the paper, that this is not how they're distinguished. But it makes the table very confusing.
  - Instead of having to refer to, e.g., TR5 3N, it would be much clearer to name the treatment like "Apr1\_60kgN."
  - Figures like Fig. 6 should have fully meaningful axes and labels. So instead of "treatment" on the X axis, have sowing date or fertilization level. And instead of "Fx#" in the titles, have actual numbers.
  - In figures like Fig. 7, treatments 1-4 are marked as Fx0, but according to Table 1, TR2 is Fx1 and TR3 is Fx2.
  - Why say things like "Fx1" when you could just say the actual amount of N applied?
- Agro-climate indicators and extremes
  - $\circ$   $\;$  These should be introduced and explained in Methods, not Results.

- What does the continentality index *mean* as far as maize is concerned?
- $\circ$   $\;$  Why are the scorching index results not in the Extremes section?
- $\circ$   $\,$  Why are the total precipitation results in the Extremes section?
- It probably would be better to separate these into subsections for temperature and precipitation, rather than "indicators" and "extremes." Because aren't the extremes also measured using indicators?
- L 359-361: In contrast to what this text says, none of these actually had significant trends.
- Experiments and analyses should be explained and justified in the Methods, with the Results section focused on actual results and some interpretation.
- What is the purpose of the analysis in Sect. 3.b.3 ("Sensitivity to changes to nutrients")? How can farmers *choose* inherent characteristics of their soil? Because the paper is rather long, every analysis should be well-justified. This one seems like it could be removed, both because its usefulness is unclear and because it distracts from the actually-interesting bit of the paper (genotype identification).
- Sect. 3.c ("Optimal genotype identification") needs a complete rewrite. It is nearly impossible to understand due to the extensive use of abbreviations; I don't have the time needed to do the deciphering necessary for a review of its content.
  - L 611-2: "the slopes of Pi variation as a function of G-ranged index"??
- L662-70: How do the two methods compare in terms of computational time? It's not sufficient to just say how good the genetic method is after a certain number of iterations.
- According to GMD guidelines, code must be associated with a DOI, e.g. with Zenodo.

**Miscellaneous comments / corrections**

- L 142-6: L 142 says it's 3 models, but then there are five listed at L 145-6.
- Are the "cultivar related coefficients" at L 156-7 the same as the "six parameters defining the genotype" at L 153?
- L 169: "Schema from Annex1"?
- L 172: What does "static" mean here?
- L173: What is NUTS3?
- L 178-186: Per GMD guidelines, subplots in a single figure should have one combined caption, and the figure should be one single image. Either combine the captions and subplots or renumber 1a → 1 and 1b → 2. (Same for Fig. 3a/b.)
- Fig. 2 (L 200-230): Text not aligned with boxes.
- Table 1 (L 233): Suggest using e.g. "Apr. 1" instead of "1.04" for dates to avoid ambiguity and confusion.
- L 239 and following: Subsections should be labeled 3.1, 3.1.1, etc. according to GMD guidelines.
- Figs. 3a, 3b (L 270-306):
  - Fig. 3a: What is H32temp?

- Fig. 3a: What is ENS? Why does the figure with that in the caption not have an associated date range?
- Fig. 3b: Why do titles say "Martonne\*1" and "Martonne\_aridity\*1"?
- Fig. 3b caption says that both rows show deltas in the right two panels ("and changes relative to it"), but neither does.
- Fig. 3b caption: What are IM and ID?
- L 313-4: "each of the three decades" conflicts with "both decades" and the fact that only two decades are shown in Fig. 4.
- Fig. 4 (L 322-55): Change subplot titles to something meaningful.
- L 371-377:
  - $\circ$   $\;$  Text refers to "Control simulations" but Fig. 5 only shows "treatments."
  - $\circ$   $\:$  Is it possible to say which of the treatments was closest to real practices?
- Fig. 5 (L 378-383):
  - Most colors are very hard to see against white.
  - Add Y-axis label and tick numbers.
  - Were the data first normalized to Z scores before correlation analysis?
- L 393-4: How does change in anthesis date affect growing season length? Wouldn't growing season length only be affected by sowing and maturity dates?
- Fig. 6 (L 398-414):
  - Do not use red and green on the same plot, as this is hard to distinguish for people with the most common color vision deficiency.
  - Why do plots only show some treatments?
  - This figure is impossible to understand without referring back to Table 1, but some thoughtful figure design would make that unnecessary.
  - Add Y axis labels.
- L 433: What is an H value?
- Fig. 7 (L 449-475): Add Y-axis labels.
- L 483-4: How exactly would richer soil lead to the model simulating slower maturity?
- L 522-3: Why increase the soil water content? This is insufficient explanation.
- Fig. 9 (L 577-89):
  - $\circ~$  Far too small, especially considering the tiny plots inside plots.
  - What is "Hmax left"?
  - Add X axis tick marks for some points between 1 and 200.
- L 834-5: Why is the disclaimer about the US Government necessary? None of the authors have US government affiliations.
- All multi-plot figures: Add subplot labels (a, b, etc.) and refer to these in the text to help readers make the connection between what you write and what the figures show.
- Most figures are unnecessarily small; please enlarge them and make sure to use a high DPI (at least 300).

Significant work is needed on language cleanup. I've listed a number of examples here, but this list is not complete.

• L 30: Should "actual" be "current"?

- L 34: "in opposite" should be "on the other hand" or "in contrast"
- L 37-8: "but emphasizes... actual climate." I'm not sure what this means.
- L 59: "9,1 milliards" should be "9.1 billion".
- L 125-6: "mainly for isolated extremes, or broad parameters' range"-?
- L 150: Delete last comma
- L 213 (Fig. 2): "Maxim" should be "Maximum"
- L 263-4: What is "([C]+10)"? Why the +10?
- L 285: "conventionality" should be "continentality".
- L 311-13: Total precipitation is listed twice.
- L 316: "evolution" implies the long-term process of speciation. Presumably this should be "development".
- L 350: "Ox axis" should be "X axis".
- L 360: "turns in opposite" should be "flips" or "reverses".
- L 382: "the S-Romania" should be "southern Romania".
- L 389-391: This is not a complete sentence.
- L 395: "slowed grain feeling"?
- L 430: Should "Harvest" here be "yield"?
- L 434: "no more valid" should be "not true" or "no longer true".
- L 481: "overestimate"?
- L 500: "astuciously," while technically a word, is extremely obscure. Try instead: "smartly," "astutely."
- L 524: Should "known" be "no"?
- L 533: Should "intra-model" be "inter-model"?
- Throughout: "decade" should be changed to "dekad" when referring to a period of ten days, to avoid potential confusion with the much more common usage of "decade" to refer to a period of ten years.
- Throughout: When referring to figure axes, "Ox" and "Oy" should be "X" and "Y".
- Throughout: Instead of H or Harvest, just say yield.

You might consider using an LLM with a prompt like "Please clean up this text from a scientific paper," feeding it one or two paragraphs at a time. However, (a) make sure to check the output yourself to ensure it didn't remove important ideas or introduce any spurious ones, and (b) make sure to have a native English speaker check the output for remaining inaccuracies.

As an example, I asked Google Gemini to clean up the text at L 310-320, and it gave me: Figure 4 presents the projected changes in extreme weather events for the Călăraşi target subregion under the RCP85 scenario compared to the historical period (Hist). The analysis focuses on three key variables during the critical maize sowing month of April: freezing days (FD), total precipitation (RR), and severe precipitation events (RR10, days with daily accumulated precipitation exceeding 10 mm).

A decreasing trend in FD is observed for both decades under the RCP85 scenario. However, it's noteworthy that the third decade may still experience intervals with even higher FD counts compared to the historical period. This late-spring blizzard phenomenon, crucial for plant evolution, has been

linked to the combined influence of Polar Jet instability and warmer Eastern Mediterranean sea surface temperatures (Caian and Andrei, 2019). As both factors are projected to intensify in a warming climate (Lelieveld et al., 2012; Shaw and Miyawaki, 2024), the region may experience a higher potential for severe spring blizzards, which could negatively impact crops and annual yields.

Although this is an improvement, there are still issues, including:

- "target subregion"—what is "target" saying here?
- "evolution" should be "development"
- Explanation of "decade" (ten-day period, not ten years) was removed.
- Gemini changed "main sowing month" to "critical sowing month," but I think "main" makes more sense. This isn't really a big issue, though, just a matter of taste.

---

## Author Response (AR1)

**REVISOR 1:**

**Dear Revisor,**

We are deeply grateful for your extremely valuable suggestions, corrections and
highly valuable scientific questions.
We are absolutely certain that this work changed fundamentally due to your input.
We are enormously grateful and we hope we answered your requests in this revised form.
Please note that section 3.3 and 4 where completely re-written.

We add to each of your question, our answer (A):

"This paper develops a modelling system for adaptation support that can clarify the optimal genotype
and cultivation management for maize cultivation under climate change in Romania. In recent years,
research into climate change impact and its adaptation has advanced highly, and it has been becoming
increasingly important to consider how to implement the findings of this research into society. The
system developed in this paper is expected to be effectively used by the related stakeholders in
Romania. However, there are some significant points that should be improved in this paper.    Firstly,
although a sensitivity analysis was carried out on various factors in relation to yield, the developed
system's predictive reality and effectiveness could not be fully convinced me due to the lack of
explanation of the details of the processes considered by the crop model and the lack of sufficient
biological background to the results. Secondly, there are many basic mistakes throughout the paper, so I
hope the authors will check it carefully."

[Specific comments]

Q: 3.a.1:How general is the index used in this study?

*A: It is an index that characterizes the impact of large-scale (temperature) on regional scale (it
accounts for the regional annual gradient, expressed by the maximal range of monthly temperature),
detailed now at lines 215-221*

Q: 3.a.2:Without an explanation of the agrometeorological background in Romania, the importance of
these indices is not understood.

*A: The choice of the three indices was explained largely in the regional context, in the revised version.
The aim was to analyse the two main climate variables implied in the region's agro-climate:
Temperature (JCI) and Precipitation (IM) , and also to represent the regional aspects, choosing a main
index considered in local agro-meteorological analysiss: the H32 Scoarching that involves total
number of degrees over 32C accumulated in JJA,  (lines: 222-230).*

Q: Fig.5: Please write the vertical axis of the graph. Though the meteorological data since 1976 were
used, from what year were yield data obtained? Please illustrate the missing years so that they are clear
(it seems that at least the years 1992, 93 and 94 are missing).

*A: the plot has been redone, following the Revisor's indication: we included missing data and specified axes. The plot is in now in Supplementary S1, lines 805-810, after we revised the document in order to considered the requirement of restraining it.*

Q: The prediction in 1995 does not seem to match the observations. What is the reason for this? The variability of the predictions seems to be greater than that of the observations. This would lead to over-underestimation in future predictions. Which of the 12 management scenarios is the closest to reality?

*A: we detailed the harvest result of 1995 in lines 286-291*

Q: L423-425: I doubt whether it can be explained by rainfall. Yield is affected by various weather factors. For example, when the photosynthesis period is shortened due to a shortened growing season, the increase in biomass may not be sufficient.
*A: Your statement helps us correcting an important issue. The "attributed" word was replaced by "reated to" at line 315.*

Q: L426-428:From Fig. 7c, it was not possible to confirm whether fertilizer efficiency decreases due to global warming. This may be due to an increase in photosynthetic products.

*A: We re-formulated that the efficiency gets lower in the simulations when coupled with climate scenarios, not excluding other causes. (lines 23, 315-320)*
*Since we compare simulations radiatively forced versus the same simulations without radiative forcing, the difference can be attributed to the difference in forcing (but indeed, this is including there all the non-linear interactions and feedback with other components). Second, this feature appears robust, found in each model of the ensemble.*

Q: 3.b.3:Is it practically possible to change N and C without adding fertilizer?
*A: Nitrogen and organic carbon contents may vary for a given type of soil not only due to fertilization from the current year, but in a much smaller extent, also due to natural heterogeneity, previous crops (especially in case leguminous crops), conservative agriculture practices, microorganism activity etc. In our case, we performed this sensitivity test aiming at the application of the system to other locations. The impact of these soil characteristics is significant and should be addressed in dedicated study.*

Q: L483-4:Why does soil fertility delay maturity?   corelat cu stresul din perioada reprodictiva.
*A: An explanation is given at 304-306 , correlated with the stress along reproductive stages.*

Q: L492-3:I can't understand the sentence.
*A: The paragraph was re-written*

Q: L516-520:Why were these six parameters chosen? What are the values of each? What is the validity of the values that were adopted?
 *A: We explained the reason (these are main parameters in the DSSAT model) and each parameter was explained as meaning, previous research and the values use din this study at lines: 131-159.*

[Minor comments]

It is better to explain what an abbreviation stands for when it first appears.
    *We have corrected, and verified in the revised version*

RCP45 and RCP85 → RCP4.5 and RCP8.5:
    *now is corrected*

Please standardize terminology (e.g. yield and harvest, sowing and planting)
    *- planting was replaced with sowing*
    *- Yield and harvest were maintained different in just few cases: yield in cases with agro-climate meaning and "harvest" mainly in cases for model output result.*

 L 28:RCP scenario is an emission scenario, not a climate scenario.
    *Was corrected*

L51:")" is needed after Hatfield et al, 2021.
    *the parenthesis was corrected*

L56:"," before Xie et al, 2023 is needed to be corrected to ";".
    *was corrected*
L319:";" after Miyawaki, 2024 is needed to be removed.
    *was corrected*
L361:peack → peak
    *was corrected*
L375:Yield → yield
    *was corrected*
L651:tis → this
    *was corrected*

Table1:I think that Fx0 is correct for Fertilazation in TR2 and 3.
    *Table 1 was corrected, lines 206-208*

Table1:I think that for the sowing date in TR2, 6 and 10 15.04 is correct.
    *Table 1 was corrected,  lines 206-208*

**REVISOR 2**

**Dear Revisor,**

We are deeply grateful for your extremely valuable suggestions, corrections and highly valuable scientific questions.
We are absolutely certain that this work changed fundamentally due to your input.
We are enormously grateful and we hope we answered your requests in this revised form.
Please note that section 3.3 and 4 where completely re-written.

Q: What are the genotype parameters that are getting modified? What do they represent in terms of processes?
o This is only explained deep into the Results section (L516-519). This should be in the Methods instead.
A: *We corrected this. There is now detailed description of parameters in the "Methods" part, lines: 131-159*

Q If P4 was kept constant, why is it even mentioned?
A: *Yes, P4 was constant. Since results are depending on the value, we specified for the reader the value used in simulations at lines 145-147.*

Q: Treatment naming is very confusing, which results in figures that are hard to understand.
A: *we corrected this, we specified the sowing date instead of treatment number in all figures.*

Q: Looking at Table 1, what is the difference between Fertilization (3N) and Fertilization (1N)? How can, e.g., TR2 get both 60 and 23 kgN/ha? I think, from reading the rest of the paper, that this is not how they're distinguished. But it makes the table very confusing.
A: *We denote two experiments: exper "1N" and exper "3N", and fertilisations Fx0, Fx1, Fx2 have values dependent on the experiment: Fx0 is no fertlisation, Fx1 is the unit fertilisation of the experiment and Fx2 is the double unit fertilisation of the experiment. We define the unit fertilisation of the exper "1N" equal to 23 N/kg and the unit fertilisation of the exper "3N" as 60 kg/ha. (lines 204-209)*

Q: Instead of having to refer to, e.g., TR5 3N, it would be much clearer to name the treatment like "Apr1_60kgN."
A: *yes, is a good possibility also. But we wanted to emphasize the ratio (e.g. no or doubling fertilization.. etc.) for easier connection with result interpretation.*

Q: Figures like Fig. 6 should have fully meaningful axes and labels. So instead of "treatment" on the X axis, have sowing date or fertilization level. And instead of "Fx#" in the titles, have actual numbers.
A: *We have corrected this as required by the Reviewer for both: tratmnet number was replaced by the sowing date; instead of "Fx#" in the titles, we have now actual numbers.*

Q: In figures like Fig. 7, treatments 1-4 are marked as Fx0, but according to Table 1,

TR2 is Fx1 and TR3 is Fx2.
A: *we corrected this error in the Table1. Treatments 1-4 have zero fertilisation (Fx0).*

Q: Why say things like "Fx1" when you could just say the actual amount of N
applied?
A: *we tried to correct and avoid along the text now. The aim of this is to point to ratios the interpretation of results, equal spacing connects easier with linearity/ non-linearity.*

• Agro-climate indicators and extremes
o These should be introduced and explained in Methods, not Results. What does the continentality index mean as far as maize is concerned?
A: *This part is an assessment section, work that has been done to to justify the necessity of the paper. To show that this is a hot-spot region of Europe (also confirmed by Copernicus 2023 report), warming more than surroundings and faster. Is not a direct part of the Methods related to main results, but rather an introduction work, justificatory. Continentality index shows us the percent and speed of change. towards a new climatic class, so the risk of seeing in the near-future a fully different shape of crop-use in the region (warmer summers in the agricultural area), linked to new climate sub-class, as stated in lines 2019-201.*

Q: Why are the scorching index results not in the Extremes section?
A: *This is an operational field, its values are used for crop characterization (there are operational thresholds), not only for extreme cases.*

Q: Why are the total precipitation results in the Extremes section?
A: *total precipitation is needed here together with severe R10mm precipitation. Only analyzing both field, together, one can conclude if it will rain less but more extreme or it will rain more and also more extreme.*

Q: It probably would be better to separate these into subsections for temperature
and precipitation, rather than "indicators" and "extremes." Because aren't the
extremes also measured using indicators?
A: *Separation was hard upon Temperature and Precipitation because there are indices involving both variables (e.g. de Martonne). Yes, indeed, extremes are measured with indicators, meanwhile also direct variables' extreme is here discussed (e.g. RR).*

Q. L 359-361: In contrast to what this text says, none of these actually had significant
trends.
A: *In dekade 3 of April, FD has significance at 5% level while RR10 and RR have significance at 10% level. This change is emphasised, with its statistical significance values, as it can be very important for future adaptation.*

Q: What is the purpose of the analysis in Sect. 3.b.3 ("Sensitivity to changes to nutrients")?
A: *This section is now moved in the supplementary material*

Q: How can farmers choose inherent characteristics of their soil? Because the paper is
rather long, every analysis should be well-justified. This one seems like it could be
removed, both because its usefulness is unclear and because it distracts from the

actually-interesting bit of the paper (genotype identification).
A: *Indeed, this section is now moved in supplementary material, shortened and focused. Its usefulness is for further portation of the pilot system over different regions.*

Q: Sect. 3.c ("Optimal genotype identification") needs a complete rewrite. It is nearly impossible to understand due to the extensive use of abbreviations; I don't have the time needed to do the deciphering necessary for a review of its content.

A. *This section has been completely re-written, re-organised, and focused*

Q: L 611-2: "the slopes of Pi variation as a function of G-ranged index"??
A: *was re-formulated in lines 466-469*

L662-70: How do the two methods compare in terms of computational time? It's not sufficient to just say how good the genetic method is after a certain number of iterations.
A. *Quantification of computational time differences is now included at lines: 477-479*

*According to GMD guidelines, code must be associated with a DOI, e.g. with Zenodo. The code was uploded on Zenodo,* DOI *10.5281/zenodo.13145522* (DSSAT code used in PREPCLIM project)

*and*
*10.5281/zenodo.13132588*
*(PREPCLIM software - additional material for publication in "Geoscientific Model Development" 2024)*

Miscellaneous comments / corrections

*) L 142-6: L 142 says it's 3 models, but then there are five listed at L 145-6.
        this was a mistake, apologises and many thanks. This is now corrected.
A: *There are 3 models discussed here (later work, after this submission, extended the number of models)*

*) Are the "cultivar related coefficients" at L 156-7 the same as the "six parameters defining the genotype" at L 153?
A: *Yes, we use this two termes for same parameters, now this is stated more clearly at line 152.*

*) L 169: "Schema from Annex1"?
A: *replaced now with "described in Annex" at lines 173-4*
*) L 172: What does "static" mean here?
A: *explained at line 176*
*) L173: What is NUTS3?
A: *explained at line 176-7*
*) L 178-186: Per GMD guidelines, subplots in a single figure should have one combined caption, and the figure should be one single image. Either combine the captions and subplots or renumber 1a → 1 and 1b → 2. (Same for Fig. 3a/b.)
A: *we corrected this aspect*

*) Table 1 (L 233): Suggest using e.g. "Apr. 1" instead of "1.04" for dates to avoid ambiguity and confusion.

*A: Indeed, this would be ideal, however there was no place to put it with high, clear size in labels in Figures, so for consistency we used same notation as in Figures.*

*) L 239 and following: Subsections should be labeled 3.1, 3.1.1, etc. according to GMD guidelines.

*A: we corrected this aspect*

*) Figs. 3a, 3b (L 270-306):
o Fig. 3a: What is H32temp?•

*A: the definition is now explained at lines 222-224 and the threshold values explained on the Fig.3 caption, lines 233-8.*

o Fig. 3a: What is ENS? Why does the figure with that in the caption not have an associated date range?

*A: "ENS" was removed from the text (used "ensemble" instead);*

o Fig. 3b: Why do titles say "Martonne*1" and "Martonne_aridity*1"?

*A: we redone the figures with uniformised tex*t
o Fig. 3b caption says that both rows show deltas in the right two panels ("and changes relative to it"), but neither does.

*A: The figures are correct;  these are indeed deltas in H32 figure (new Fig 3b), we re-verified the data and the plots, and specified also in the text: the index almost doubles under climate scenarios (line 224).*

*We attach here the plots with full field H32 [C] for Historical (left, 1970-2000), RCP4.5 (middle, 2021-2050) and RCP8.5 (right, 2021-2050)*

[Figure]

[Figure]

[Figure]

o Fig. 3b caption: What are IM and ID?

*A: JCI defined line 2015; IM define line 266; ID is no more used.*

*) L 313-4: "each of the three decades" conflicts with "both decades" and the fact that only two decades are shown in Fig. 4.

*A: corrected, there is no more "three decades" in the text.*
Fig. 4 (L 322-55): Change subplot titles to something meaningful.

*A: subplot titles were uniformised.*

*) L 371-377:
o Text refers to "Control simulations" but Fig. 5 only shows "treatments."

*A: Control simulations (Ctrl) were defined at lines 280-1 (ERA5 driven).*

Fig.5 was moved to supplementary material, S1.

o Is it possible to say which of the treatments was closest to real practices?
*Fig. 5 (L 378-383):*
*A: in every year, another treatment was closer but in specific cases where data were available, we verified against data from the Official Data basis, as the case of 1995 year, and obtained that if conditions are closer to the used ones, the results are the closest to measurements (line 289-291).*

o Most colors are very hard to see against white.
*A: the figure was re-done and moved to supplementary S1)*
o Add Y-axis label and tick numbers.
*A: the Y-axis label was redone, figures are bold (original Figure, more clear, is included in Figure's tar, in the text it might be a problem of import file)*
o Were the data first normalized to Z scores before correlation analysis?
*A: no normalisation was applied, nor bias correction (we added this specification in S1)*

L 393-4: How does change in anthesis date affect growing season length? Wouldn't growing season length only be affected by sowing and maturity dates?

*A: In the DSSAT MODELS the anthesis and maturity dates are calculated (and not pre-defined), for each simulation based on thermal time, and cultivar dependent parameters. As you mention, anthesis date change, cannot **alone** lead to season's length change. The stress factors for water, temperature and nitrogen may impose a premature ending of grain filling or crop failure.*

Fig. 6 (L 398-414):
o Do not use red and green on the same plot, as this is hard to distinguish for people with the most common color vision deficiency.
        A: *Solved (the colors are now red, blue and black)*

o Why do plots only show some treatments?
        A: *Explained in line 305: only small differences appear (linked to model instability under enhanced fertilization) in the maturity dates.*

o This figure is impossible to understand without referring back to Table 1, but some thoughtful figure design would make that unnecessary.
        A: *Additional explanations were added in the legend of Fig. 6*

o Add Y axis labels.
*A: solved*
L 433: What is an H value?
        A: *Shortcut for harvest is defined as H at line 316*

Fig. 7 (L 449-475): Add Y-axis labels.
A: *Solved*
L 483-4: How exactly would richer soil lead to the model simulating slower maturity?

A: *"In our study this premature ending of simulated vegetation season occurred more frequently in treatments with higher nitrogen fertilization. This may favour leaves development, enhanced transpiration and earlier depletion of the soil moisture leading later to water stress."* line 304-306

L 522-3: Why increase the soil water content? This is insufficient explanation.

A: *We increased the initial soil water content by 5% as indicated by the projected maximum change over the pilot area,(genotyping is an adaptation Tool so getting closer to projected conditions could bring more accuracy)*

Fig. 9 (L 577-89):
o Far too small, especially considering the tiny plots inside plots.
      A: *Inside plots were removed, figure enlarged, other changes available at editors requests (line 390)*

o What is "Hmax left"?
      A: *shows the sense of the axis: increasing values of H are on leftwards direction of the axis (line 394).*

o Add X axis tick marks for some points between 1 and 200.
*A: solved*

L 834-5: Why is the disclaimer about the US Government necessary? None of the
authors have US government aSiliations.
A: *removed*

Q: All multi-plot figures: Add subplot labels (a, b, etc.) and refer to these in the text to help
readers make the connection between what you write and what the figures show.
Most figures are unnecessarily small; please enlarge them and make sure to use a high
DPI (at least 300).
A: *Figures were enlarged, zooms were removed*

Q: Significant work is needed on language cleanup. I've listed a number of examples here, but this list
is not complete.
• L 30: Should "actual" be "current"?•   A: *Replaced*
L 34: "in opposite" should be "on the other hand" or "in contrast" "in opposite" was replace with "in
contrast"
L 37-8: "but emphasizes... actual climate." I'm not sure what this means. (no more occurring)
L 59: "9,1 milliards" should be "9.1 billion". A: *corrected at line 44.*
L 125-6: "mainly for isolated extremes, or broad parameters' range"—?
A: *re-written, this does no more occur*
L 150: Delete last comma. A: *whole sections 3.3 and 4 were rewritten*
L 213 (Fig. 2): "Maxim" should be "Maximum": A: *corrected*
L 263-4: What is "([C]+10)"? Why the +10?    A: *this is the standard formula for the indicator.*
L 285: "conventionality" should be "continentality".   A: *corrected*
L 311-13: Total precipitation is listed twice.  A: *corrected*
L 316: "evolution" implies the long-term process of speciation. Presumably this should
be "development".    A: *indeed, but present in ML literature for the "evolution" of a "population" in GA/ ML. Only associated with GA we maintained "evolution" in the text.*

L 350: "Ox axis" should be "X axis". A: *corrected*

L 360: "turns in opposite" should be "flips" or "reverses". *A: was rewritten, does no more occur*

L 382: "the S-Romania" should be "southern Romania". A: *Corrected*

L 389-391: This is not a complete sentence. A: *The paragraph was rewritten*

L 395: "slowed grain feeling"? A: *The paragraph was rewritten*

L 430: Should "Harvest" here be "yield"? A: *We used both, harvest mainly related to model values (in the 3.3 section)*

L 434: "no more valid" should be "not true" or "no longer true". *A: rewritten, does no more occur*

L 481: "overestimate"? A: *re-written, does no more occur*

L 500: "astuciously," while technically a word, is extremely obscure. Try instead: "smartly," "astutely." A: *re-written, does no more occur*

L 524: Should "known" be "no"? A: *re-written, does no more occur*

L 533: Should "intra-model" be "inter-model"? A: *re-written, does no more occur*

Throughout: "decade" should be changed to "dekad" when referring to a period of ten days, to avoid potential confusion with the much more common usage of "decade" to refer to a period of ten years. A: *this change has been done ("decade" was replaced with "dekad")*

Throughout: When referring to figure axes, "Ox" and "Oy" should be "X" and "Y". A: Corrected

Throughout: Instead of H or Harvest, just say yield.

A: *We used both, harvest mainly related to model values (in the 3.3 section), mainly when we needed to avoid repetition in the same phrase.*

Q: You might consider using an LLM with a prompt like "Please clean up this text from a scientific paper," feeding it one or two paragraphs at a time. However, (a) make sure to check the output yourself to ensure it didn't remove important ideas or introduce any spurious ones, and (b) make sure to have a native English speaker check the output for remaining inaccuracies.

A: *We have followed Revisor advice, we considered LLM on the text, and adopted tool's suggestions, in parts of the text. We are grateful for the suggestion, indeed.*

As an example, I asked Google Gemini to clean up the text at L 310-320, and it gave me:

Figure 4 presents the projected changes in extreme weather events for the Călăraşi target subregion under the RCP85 scenario compared to the historical period (Hist). The analysis focuses on three key variables during the critical maize sowing month of April: freezing days (FD), total precipitation (RR), and severe precipitation events (RR10, days with daily accumulated precipitation exceeding 10 mm). A decreasing trend in FD is observed for both decades under the RCP85 scenario. However, it's noteworthy that the third decade may still experience intervals with even higher FD counts compared to the historical period. This late-spring blizzard phenomenon, crucial for plant evolution, has beenlinked to the combined influence of Polar Jet instability and warmer Eastern Mediterranean sea surface temperatures (Caian and Andrei, 2019). As both factors are projected to intensify in a warming climate (Lelieveld et al., 2012; Shaw and Miyawaki, 2024), the region may experience a higher potential for severe spring blizzards, which could negatively impact crops and annual yields.

Although this is an improvement, there are still issues, including:
• "target subregion"—what is "target" saying here? A: *target area defined at line 200.*
• "evolution" should be "development". A: *replaced, apart GA*
• Explanation of "decade" (ten-day period, not ten years) *was removed*.
• Gemini changed "main sowing month" to "critical sowing month," but I think "main"

makes more sense.

---

## Referee Report (RR1)

**Re-review: "A modeling System for Identification of Maize Ideotypes, optimal sowing dates and nitrogen fertilization under climate change - PREPCLIM-v1" (gmd-2024-105)**

Unfortunately, the authors' revisions did not do much to improve the paper's organization, language, or figures, which were the three major themes of my first review. I recommend another set of major revisions.

The issues of most critical importance to the paper are marked in **bold**.

General

1. Are these tools publicly accessible? If so, please provide URLs. If not, please explain why.
2. **Figures throughout (including the Supplement) are very low-quality with obvious JPEG artifacts. PDF should be used when possible for vector-based figures and PNG elsewhere, with a resolution of at least 300 dpi. (JPEGs should only ever be used for photographs.) See "Figure composition" bullet at [https://www.geoscientific-model-development.net/submission.html#figurestables](https://www.geoscientific-model-development.net/submission.html#figurestables)**
3. **Code is still not associated with a DOI, despite the GMD requirement: [https://www.geoscientific-model-development.net/policies/code_and_data_policy.html#item3](https://www.geoscientific-model-development.net/policies/code_and_data_policy.html#item3)**

Abstract:

4. L18: Specify *Southern* Romania.

Sect. 1: Introduction

5. L90: What is a "cross-range"?
6. **L110: Portability is more than just showing that changing inputs doesn't change the results much, which seems to be what Sect. S2 is saying, although it's very unclear. I suggest deleting this sentence, as well as deleting Sect. S2, which is an unnecessary hodgepodge of manipulations that don't seem comprehensive enough to draw meaningful conclusions from. It's just distracting and confusing.**

Sect. 2: Data and Methods

7. Split Sect. 2 (Data and Methods) into subsections for science (L119-173) vs. software (L174-204).

8. **From reading Sect. 2 (Data & Methods), I don't have a sense of whether the optimal management and cultivars are allowed to evolve over time. Is the optimization taking place for each year?**

9. L137-140: This description of P2 is hard to understand. What does it mean to "delay" development? Can P2 be summed up as, "Longer days increase plant growth only up to a point P2, above which plant growth decreases"? If so, please explain why.

10. I ask again: If P4 was kept constant, why is it even mentioned? You only analyze responses across five parameters, so why talk about this sixth one? Is it because it's something that the application COULD analyze, you just didn't do it here? That's relevant for the software side of things but not the science.

11. L149: Thermal time parameter is missing (a) base temperature and (b) and time component. Is it 3-70 °C-*days*? Above what base temperature?

12. L154: "representatives" should be "representativeness".

13. **L154-155: What did you actually do to "rigorously test" the parameter range? What "analysis of extreme values"? If you mention these tests/analyses, you need to give details of their methods and results.**

14. **L155-6 and throughout the rest of the manuscript: For clarity, do not say "Pi" when you can just say "parameter" or "parameters" instead.**

15. L164: It's not a "proposed" approach; it's the approach you actually used. Delete "proposed".

16. L174: "optimal paths" of what? Cultivars and management?

17. L175:
   a. "one-way interactive (static)" confuses more than it helps. Please consider deleting, because "providing agro-climate information" already implies "the user is just browsing existing content, not generating anything themselves."
   b. Mention that NUTS3 in Romania corresponds mostly to the county level.

18. L177:
   c. "climate -agro-climate" typo?
   d. What indicators and indices?

19. L204-208 (Table 1 caption) and elsewhere throughout paper: Replace "exper" with "experiment."

20. Table 1 is not mentioned anywhere in its section.

21. **Table 1 is still extremely confusing.**
   e. **The authors now explain that "1N" and "3N" are experiments, but they don't explain *why* they're experiments. The text in Sect. 2 says at L159-160, "By default, the twelve agro-management scenarios encompass**

four sowing dates (spaced five days apart) and three fertilization levels (zero, then a regional average and its double).” That explains *either* 0-60-120 (“3N”) or 0-23-46 (“1N”), but I don’t understand why the authors have *both*. What exactly is the regional average? Is it 23 or 60?

    f. It’s very confusing to have one “treatment,” e.g. TR7, corresponding to both “May 5 planting with 60 kgN/ha” and “May 5 planting with 23 kgN/ha.” Why are those not designated as separate treatments within a single experiment?

Sect. 3: Results

22. It’s still very jarring to see the agro-climatic indicators introduced in a Results section. The authors’ explanation that this section is simply to “justify” the work makes it even odder—generally those kinds of things are in a Methods section titled something like “Study Region.” This paper is about the experiments and the software; the region the authors chose to test is of secondary importance. The authors’ citation of the Copernicus 2023 report confirming that the region is a European hotspot further confuses me—why include this three-page analysis, with climatic indicators that the reader is almost certainly not familiar with and which haven’t been previously explained? I *strongly* suggest the authors (a) add a subsection at the beginning of Sect. 2 titled something like “Study Region” consisting of a paragraph or two describing how the region is a hotspot of climate change but not introducing any original analysis. The authors’ analyses can be included in a Supplement instead, so as not to distract from the focus of the paper. This will also allow me to be less critical of the organization of the authors’ analyses, since the separation into “indicators” vs. “extremes” is still giving me trouble (although the authors did explain well why my “temperature” vs. “precipitation” idea wouldn’t work). It would also make it perfectly fine to have the indicators explained in the midst of their results—indeed, this would work better! Any tidbits from the authors’ analyses that are especially interesting and/or useful for interpreting results can be mentioned in the new Methods subsection, with reference made to the new Supplement section.

23. L213: Again, specify that NUTS3 in Romania mostly corresponds to the county level.

24. Fig. 5:

    a. In addition to “NUTS region 103032,” say the name of the place.

b. Needs in-figure legend explaining the lines, their colors, and what the shading represents.

c. Y-axis labels needed with text explanations and units

25. L269: No significant or near-significant decreasing trend is observed in the first dekad for either RR10 (p=0.7, Fig. 5b left side) or RR (p=0.3, Fig. 5c left side).

26. L279: Section 3.c?

27. **L280-284: Model validation needs its own subsections in the methods and at the beginning of the results. While three pages are dedicated to what is essentially a supplementary analysis (agro-climatic indicators/extremes), in this revision the validation of the model that is the *actual focus of the paper* only gets two sentences (L280-284), including one for the methodology (in the Results section for some reason), and its results figure is shunted off to the Supplement. This is a critically important part of the paper and *must* be treated as such.**

28. L286-291: Speculation about how models could be improved is material for a Discussion section, not Results. Also, where do the authors get the data about 1995's real values being close to 80-120 kgN/ha and April 15th?

29. Figs. 6 and 7:

d. What is "ENS"? Ensemble? Ensemble of what? Does each data point represent an ensemble mean? If so, uncertainty intervals should be added.

e. Need in-figure legend explaining the colors. From the GMD guidelines at https://www.geoscientific-model-development.net/submission.html#figurestables: "A legend should clarify all symbols used and should appear in the figure itself, rather than verbal explanations in the captions."

30. L301-306, 316-317, 325-326: These results should be illustrated with figures (supplement OK). Also, what was the method for the correlation analyses?

31. L319: "H difference Hist minus scenario"?

32. Fig. 8: Needs in-figure legend explaining the colors. From the GMD guidelines at https://www.geoscientific-model-development.net/submission.html#figurestables: "A legend should clarify all symbols used and should appear in the figure itself, rather than verbal explanations in the captions."

33. **L337-479 (Sect. 3.3):**

f. **Instead of GX and GI, refer to these percentile ranges as "upper"/"top" and "middle"/"intermediate". Also, why is the intermediate range 25th-70th (asymmetric around median) rather than 25th-75th?**

g. **Again, avoid the use of things like Pi and P0i, which make this section hard to parse. Use words instead.**

34. L372: Why are some numbers in parentheses?

**35. L380-385: I don't understand this almost at all.**

36. Fig. 9:

  h.   Legends should have sowing date + fertilization level instead of TR#.

  i.   What is ORD?

  j.   All the text about Fig. 9 refers to percentile ranges, so those should be the X axis, not rank. Specifically be sure to mark the 2.5th, 25th, and 70th percentiles, labeling ranges GI and GX.

  k.   Each one of these lines is an ensemble across three climate models, right? What is the inter-model variation like?

  l.   Fig. 9a: What is the arrow?

  m.   Why are lines in Figs. 9b and 9c not monotonically increasing?

37. L415-417: Please include P# labels here for ease of comparing the text to the figure.

38. L418: What are the "main stages of the development"?

39. Fig. 10:

  **n.   Too small.**

  **o.   I don't understand what the X axes are supposed to be here.**

  p.   Where is Harvest?

  q.   Needs in-figure legend explaining the colors. From the GMD guidelines at [https://www.geoscientific-model-development.net/submission.html#figurestables](https://www.geoscientific-model-development.net/submission.html#figurestables): "A legend should clarify all symbols used and should appear in the figure itself, rather than verbal explanations in the captions."

  r.   What are the things in the background? Full ensemble ranges for red and black lines? Why not also blue?

40. Fig. 11:

  **s.   Too small.**

  **t.   I don't understand what the X axes are supposed to be here.**

  u.   Where is Harvest?

  v.   Needs in-figure legend explaining the colors. From the GMD guidelines at [https://www.geoscientific-model-development.net/submission.html#figurestables](https://www.geoscientific-model-development.net/submission.html#figurestables): "A legend should clarify all symbols used and should appear in the figure itself, rather than verbal explanations in the captions."

**41. L 462-469: I don't understand this at all.**

"Annex" (should be "Appendix" in *GMD*'s style):

42. Please number the steps.

43. L805: Repeat starting from which step?
44. Consider putting this in Sect. 2 (Data and Methods), because that section is rather short anyway, and *GMD* encourages technical details.

Supplement:

45. **All figures: Do not use red and green in the same figure, as this is difficult for people with the most common form of color-blindness. See yellow box at the top of [https://www.geoscientific-model-development.net/submission.html#figurestables](https://www.geoscientific-model-development.net/submission.html#figurestables)**
46. Fig. S1:
    a. **Move back to main text (see above).**
    b. **Use date + fertilization instead of TRT #.**
    c. What are the four-digit numbers? The observed values? Why include these?
    d. Many colors are hard to see against the white background.
    e. Missing values should be represented as breaks in the lines rather than zero.
47. **Sect. S2: Just delete this; see comment about L110 above.**
48. Fig. S3:
    f. **Too small.**
    g. Why here do you split into 1-200 and 201-1890 as opposed to the percentile ranges from the main text?
49. **Fig. S4 is so small, and the image quality is so low, that the figure is unintelligible.**

---

## Referee Report (RR2)

**Re-review 2: "A modeling System for Identification of Maize Ideotypes, optimal sowing dates and nitrogen fertilization under climate change - PREPCLIM-v1" (gmd-2024-105)**

The authors have significantly improved this manuscript, but some issues still remain. I think the editor can handle them, though, so I'm happy to say accept after minor revisions.

**General**

1) Thank you for adding the link to the info platform; I was able to access it successfully. (That's a very nice interface!) Please also add the link to the Code Availability section.
2) Thank you for adding code/data DOIs. Please also mention them in the Code and Data Availability sections.
3) Figures are still too low-resolution, and this results in some of them being hard to read. The authors mention that they're now 1000x800 pixels, but that by itself doesn't mean anything—what matters is the pixels *per inch*. The GMD submission guidelines specify at least 300 ppi. Fig. 8, for example, is conservatively about 6 inches wide. That would require it to be at least 300 x 6 = 1800 pixels across, not 1000. Note that using PDF instead of PNG would be preferable for most of these plots, as—being a vector-based format—it allows "infinite resolution." Fig. 8 might be an exception, because the large number of objects (background points) would cause the PDF to be extremely large.

**Sect. 2: Data and Methods**

4) New text describing P2 in the tracked-changes version (L150-152) seems not to have made it into the final manuscript. Was this intentional? I find the sentence after "Or:" to be a helpful description.

**Sect. 3: Results**

5) L280: Replace "exper" with "experiment".
6) L294-297 (Fig. 4 caption):
   a) Replace "exper" with "experiment".
   b) Mention that the lines represent the mean for each treatment x climate.
7) L286-291: "Under warmer climates we note more frequent occurrences of critical situations with suboptimal grain filling and potential crop failure, under fertilization…. In our study premature ending of simulated vegetation season occurred more frequently in treatments with higher nitrogen fertilization, leading in average only small changes in maturity days." Reiterating my request that this be illustrated with a

figure (in the Supplement is okay). The authors provided one in their response (albeit without a legend or Y-axis labels) but seemingly not in the manuscript or Supplement.

8) L303: "in the Ctrl and in model simulations": Replace "model" with "future". They're all *model* simulations.

9) L305-308: When I first re-read this, I thought "they should really look at the changes in extremes, too"—which you do later. Consider saying here something like "Further analysis on the change in intermediate and extreme harvest values can be found in Sect. 3.3.1," or reorganizing to put these analyses together.

10) L311-313: "The correlation along sowing dates between H and accumulated precipitation until maturity (Pmat, Fig.6), is r(H, Pmat) >0.96 in both scenarios." Fig. 6 does suggest this, but showing the dots for each ensemble member would help. It would match the text I quoted even better to draw the best-fit lines for each point cloud rather than a line through their means at each date. Also, the X-axis (and independent variable in the correlation tests) should be the quantitative sowing date rather than the categorical "treatment"—these are almost identically-spaced but not exactly.

11) L354: Upper limit of "intermediate" interval still says 70% here, whereas in their response the authors say they changed it to 75%.

12) L355: Replace "projected higher H values in GI" with "projected higher intermediate H values".

13) L369-375:
   a) L371-372: Replace "GI and also in GX" with "both the intermediate and top percentiles".
   b) L374: "leading"?
   c) Mention any interesting results from Fig. S3. E.g., how some ensemble members actually show improvements. Also, it seems like there is substantial inter-model spread in the historical and to a lesser extent future periods—this is not a dealbreaker but it should be acknowledged in the main text.

14) Fig. 7a
   a) This is actually harder to read in some ways than it was before (then Fig. 9a). The first two lines for each color are impossible to distinguish. Using a PDF as requested in the guidelines could help with this.
   b) Fig. S3 should be mentioned in the Fig. 7a caption.
   c) I still don't understand what I'm supposed to take away from the "mitigation window." How was it drawn, exactly? I.e., how were the edges determined?

15) Figs. 7b and 7c: Y-axis labels should be "Difference in harvest relative to Hist".

16) Fig. 8 is still too low-resolution to properly display the point clouds in the background.

17) L423: "Ox" should be changed to "X axis".

18) L452: "Ox" should be changed to "X axis".
19) L459: "lead" should be "lead to".
20) L462-466: I still don't understand this part (although the preceding part of this section is better than before; my previous comment 41). I don't really get what "expectancy" means, and I don't know what the Y-axis in Fig. S5 is. I think "border" at L464 should be "broader"?

**Sect. 4: Discussions**

21) L500: Leftover "P0i" that didn't get changed.
22) L544: Leftover "P01" that didn't get changed.
23) L545: Leftover "P0i" that didn't get changed.

**Supplement:**

24) Throughout: Start section headers with "Section" and figure captions with "Figure" to clearly distinguish them each other.
25) Fig. S.1b (previously Fig. 5): The authors mention that my comments 24b ("Needs in-figure legend explaining the lines, their colors, and what the shading represents") and 24c ("Y-axis labels needed with text explanations and units") were addressed, but it doesn't look like those changes made it into the updated Supplement file.
26) L19: "Fertlization" typo
27) L26: "string" should be "strong".
28) L28: "dt-line" should be "dot-dash line".
29) Fig. S3:
   a) Refer back to Fig. 7a in the caption.
   b) These figures should also be remade to match the design of Fig. 7a.
30) Fig. S5:
   a) What is the Y-axis here?
   b) Why split at the 200th parameter instead of percentiles as in the rest of the paper?
   c) Linear regression doesn't look like a good fit for the two plots on the left.
31) L59: Replace "TR12" with the actual details of the treatment.
32) L60: Delete "GI".

---

## Author Response (AR2)

REVIEWER 1:

From what year is yield data available? This study used weather data from 1976, so yields can be compared for periods prior to 1990.

A: There are serious concerns that agricultural statistical data before the Romanian Revolution from December 1989 may be seriously biased by political influences, and anyway there were massive changes in the agro-technology after the restitution of the agricultural land of Agricultural Production Cooperatives ("CAP") and State Agricultural Enterprises ("IAS") towards the owners from 1945 and their heirs, practically begun before the application of Law 18 19/02/1991. The excessive fragmentation of agricultural land was partially and gradually mitigated through leasing and purchase, and the acquisition of modern agricultural machinery was subsequently supported by bank loans and EU funds.

Why do the 1995 estimation values differ from the observed values? This is useful information for readers in terms of understanding the limitations of model predictions.

A. That year may be regarded as a transition year. According to personal communication from older researchers there were several influences not considered by the DSSAT models (failure in weed and pest control). The estimations of FAOSTAT doesn't show major variations of the average nitrogen dose per hectare for all crops in Romania in 1995 (Figure 1) compared with 1994 and 1996, but, there is a statistical reference indicating that in Calarasi county the number of chemical fertilizer spreaders (252) was seriously reduced (with around 46%) in 1995 (Figure 2), and this should decrease the capacity of applying fertilization in the optimal period or even the application of treatments in several farms . Due to impossibility of benefiting from the optimal fertilization period, treatments with larger quantities of fertilizers (Figure 3). were probably applied to more  crops that otherwise usually are not fertilized in the South -Eastern Romania resulting in a larger fertilized area in 1995. New machinery was acquired after 1995 replacing the obsolete, worn-out devices.

[Figure]

*Figure 1 FAOSTAT estimated values of nitrogen/ha doses used in Romania between 1990 and 2003 (https://www.fao.org/faostat/en/#data/RFN)*

[Figure]

*Figure 2 Dynamics of chemical fertilizer spreaders at national level and Calarasi county of Romania (source National Statistics Institute, http://statistici.insse.ro:8077/tempo-online/#/pages/tables/insse-table)*

[Figure]

*Figure 3 Dynamics of the area of land where chemical and natural fertilizers were applied in Calarasi county of Romania (source National Statistics Institute, http://statistici.insse.ro:8077/tempo-online/#/pages/tables/insse-table)*

A 35% decrease in mechanical sprayers and dusters active in Calarasi county in 1995 as compared with 1994,(Print screen 3) and this may be related to an unfavorable pest and disease evolution. This decreasing trend of plant protection machinery continued till 2004, but the new equipment from the private sector was more performant.

*Figure 4 Dynamics of mechanical sprayers and dusters in Calarasi county of Romania between 1990 and 2005 (source National Statistics Institute, http://statistici.insse.ro:8077/tempo-online/#/pages/tables/insse-table)*

Also, which of the 12 management scenarios is closest to reality?

The 0-60-120 is relevant for many years of the historical period. The low input agrotechnology for rainfed maize was a direction preferred for the sensitivity part of the study due to economic concerns; projection simulations are using the current 0-60-120 N fertilization.

**REVIEWER 2**

**Re-review: "A modeling System for Identification of Maize Ideotypes,**

**optimal sowing dates and nitrogen fertilization under climate change -**

**PREPCLIM-v1" (gmd-2024-105)**

Unfortunately, the authors' revisions did not do much to improve the paper's organization,

language, or figures, which were the three major themes of my first review. I recommend

another set of major revisions.

The issues of most critical importance to the paper are marked in **bold**.

General

1. Are these tools publicly accessible? If so, please provide URLs. If not, please explain

why.

A01. Info-Platform is publicly available < https://climatologis.shinyapps.io/PrepClim/ > [L217]. The access to User-Platform hosted on an internal server is granted at request addressed to the correspondent author [L220].

**2. Figures throughout (including the Supplement) are very low-quality with**

**obvious JPEG artifacts. PDF should be used when possible for vector-based**

**figures and PNG elsewhere, with a resolution of at least 300 dpi. (JPEGs should**

**only ever be used for photographs.) See "Figure composition" bullet at**

**https://www.geoscientific-modeldevelopment.**

**net/submission.html#figurestables**

A02 Graphs are now in PNG format, enhanced resolution x1000, y 800. The simultaneous use of red and green colors was avoided.

**3. Code is still not associated with a DOI, despite the GMD requirement:**

**https://www.geoscientific-modeldevelopment.**

**net/policies/code_and_data_policy.html#item3**

A03 The DSSAT code used in PREPCLIM project, the PREPCLIM software and a PREPCLIM sample data set are available On ZENODO (DOI 10.5281/zenodo.13145521, DOI 10.5281/zenodo.13132587 and respective DOI 10.5281/zenodo.13133107) [L226]

Abstract:

4. L18: Specify *Southern* Romania.

A04 Done [L18]

Sect. 1: Introduction

5. L90: What is a "cross-range"?

A05 changed with "multiple parameter range" [L90]

**6. L110: Portability is more than just showing that changing inputs doesn't change**

**the results much, which seems to be what Sect. S2 is saying, although it's very**

**unclear. I suggest deleting this sentence, as well as deleting Sect. S2, which is**

**an unnecessary hodgepodge of manipulations that don't seem comprehensive**

**enough to draw meaningful conclusions from. It's just distracting and**

**confusing.**

A06 Suggestion applied (Deleted phrase)

Sect. 2: Data and Methods

7. Split Sect. 2 (Data and Methods) into subsections for science (L119-173) vs.

software (L174-204).

A07 Suggestion applied

**8. From reading Sect. 2 (Data & Methods), I don't have a sense of whether the**

**optimal management and cultivars are allowed to evolve over time. Is the**

**optimization taking place for each year?**

A08 Yes, it takes place each simulated year. [L177]

9. L137-140: This description of P2 is hard to understand. What does it mean to "delay"

development? Can P2 be summed up as, "Longer days increase plant growth only

up to a point P2, above which plant growth decreases"? If so, please explain why.

A09 Genetically some cultivars present, in different degrees, a slower phenogical advancement to flowering when the period with light during day exceed a certain value (long day plants).The process is controlled by phytochrome, that presents two reversible conformations (Pr and Pfr) which absorb red light (R) and respectively far-red light (FR). This part of the text was anyway rephrased. [L143]

10. I ask again: If P4 was kept constant, why is it even mentioned? You only analyze

responses across five parameters, so why talk about this sixth one? Is it because it's

something that the application COULD analyze, you just didn't do it here? That's

relevant for the software side of things but not the science.

A10 Suggestion applied, text referred to this parameter were removed.

11. L149: Thermal time parameter is missing (a) base temperature and (b) and time

component. Is it 3-70 °C-*days*? Above what base temperature?

A11 Base temperature is 8°C, it is mentioned at L140

12. L154: "representatives" should be "representativeness".

A12 Text rephased [L161].

**13. L154-155: What did you actually do to "rigorously test" the parameter range?**

**What "analysis of extreme values"? If you mention these tests/analyses, you**

**need to give details of their methods and results.**

**A13** Text rephased [L161].

**14. L155-6 and throughout the rest of the manuscript: For clarity, do not say "Pi"**

**when you can just say "parameter" or "parameters" instead.**

A14 Suggestion applied

15. L164: It's not a "proposed" approach; it's the approach you actually used. Delete

"proposed".

A15 Suggestion applied

16. L174: "optimal paths" of what? Cultivars and management?

A16 Suggestion applied, added "in various climate and management scenarios"

17. L175:

a. "one-way interactive (static)" confuses more than it helps. Please consider

deleting, because "providing agro-climate information" already implies "the

user is just browsing existing content, not generating anything themselves."

A17a Suggestion applied

b. Mention that NUTS3 in Romania corresponds mostly to the county level.

A17a Suggestion applied, "NUTS3 level, aligned with the European Union's Nomenclature of Territorial Units for Statistics, primarily corresponding to county level in Romania" [L212]

18. L177:

c. "climate -agro-climate" typo?

A18 c Error removed

d. What indicators and indices?

Done L212-216

19. L204-208 (Table 1 caption) and elsewhere throughout paper: Replace "exper" with

"experiment."

Done

20. Table 1 is not mentioned anywhere in its section.

Done, L164

**21. Table 1 is still extremely confusing.**

**e. The authors now explain that "1N" and "3N" are experiments, but they**

**don't explain *why* they're experiments. The text in Sect. 2 says at L159-**

**160, "By default, the twelve agro-management scenarios encompass**

**four sowing dates (spaced five days apart) and three fertilization levels**

**(zero, then a regional average and its double)." That explains *either* 0-60-**

**120 ("3N") or 0-23-46 ("1N"), but I don't understand why the authors have**

**both. What exactly is the regional average? Is it 23 or 60?**

A21 The 0-60-120 is relevant for many years of the historical period. The low input agrotechnology for rainfed maize was a direction preferred for the sensitivity part of the study due to economic concerns; projection simulations are using the current 0-60-120 N fertilization.

**f. It's very confusing to have one "treatment," e.g. TR7, corresponding to**

**both "May 5 planting with 60 kgN/ha" and "May 5 planting with 23**

**kgN/ha." Why are those not designated as separate treatments within a**

**single experiment?**

**A21 f Treatment were renamed (Table 1)**

Sect. 3: Results

**22. It's still very jarring to see the agro-climatic indicators introduced in a Results**

**section. The authors' explanation that this section is simply to "justify" the**

**work makes it even odder—generally those kinds of things are in a Methods**

**section titled something like "Study Region." This paper is about the**

**experiments and the software; the region the authors chose to test is of**

**secondary importance. The authors' citation of the Copernicus 2023 report**

**confirming that the region is a European hotspot further confuses me—why**

**include this three-page analysis, with climatic indicators that the reader is**

**almost certainly not familiar with and which haven't been previously explained?**

**I *strongly* suggest the authors (a) add a subsection at the beginning of Sect. 2**

**titled something like "Study Region" consisting of a paragraph or two describing**

**how the region is a hotspot of climate change but not introducing any original**

**analysis. The authors' analyses can be included in a Supplement instead, so as**

**not to distract from the focus of the paper. This will also allow me to be less**

**critical of the organization of the authors' analyses, since the separation into**

**"indicators" vs. "extremes" is still giving me trouble (although the authors did**

**explain well why my "temperature" vs. "precipitation" idea wouldn't work). It**

**would also make it perfectly fine to have the indicators explained in the midst**

**of their results—indeed, this would work better! Any tidbits from the authors'**

**analyses that are especially interesting and/or useful for interpreting results**

**can be mentioned in the new Methods subsection, with reference made to the**

**new Supplement section.**

A22 We took your suggestion and agro-climatic part was significantly reduced and moved to Methods and to Supplementary

23. L213: Again, specify that NUTS3 in Romania mostly corresponds to the county level.

A23 Already specified at first occurrence [L212]

24. Fig. 5:

a. In addition to "NUTS region 103032," say the name of the place.

24 a Added Ilfov county [L800]

b. Needs in-figure legend explaining the lines, their colors, and what the

shading represents.

Done

c. Y-axis labels needed with text explanations and units

Done

25. L269: No significant or near-significant decreasing trend is observed in the first

dekad for either RR10 (p=0.7, Fig. 5b left side) or RR (p=0.3, Fig. 5c left side).

A25 We kept only statistically significant results , Supplement 1.

26. L279: Section 3.c?

Done

**27. L280-284: Model validation needs its own subsections in the methods and at**

**the beginning of the results. While three pages are dedicated to what is**

**essentially a supplementary analysis (agro-climatic indicators/extremes), in**

**this revision the validation of the model that is the *actual focus of the paper***

**only gets two sentences (L280-284), including one for the methodology (in the**

**Results section for some reason), and its results figure is shunted oj to the**

**Supplement. This is a critically important part of the paper and *must* be treated**

**as such.**

A27 Validation part was moved in "3.1 Model validation"

28. L286-291: Speculation about how models could be improved is material for a

Discussion section, not Results. Also, where do the authors get the data about

1995's real values being close to 80-120 kgN/ha and April 15th?

A28 The maize yield of year 1995 in Calarasi county from the statistical  was rather close to a lower fertilization level (Supplement 2). Model improvement discussion was removed.

29. Figs. 6 and 7:

d. What is "ENS"? Ensemble? Ensemble of what? Does each data point

represent an ensemble mean? If so, uncertainty intervals should be added.

Ensemble Max and min values of the members are now plotted on the maps together with mean ensemble values.

e. Need in-figure legend explaining the colors. From the GMD guidelines at

https://www.geoscientific-modeldevelopment.

net/submission.html#figurestables: "A legend should clarify all

symbols used and should appear in the figure itself, rather than verbal

explanations in the captions." Suggestion applied

30. L301-306, 316-317, 325-326: These results should be illustrated with figures

"..changes in maturity days" due to model failure as a function of fertilization Fig. below shows for 2 fertilizations (left Fx1; right Fx2 the grain filling season length, maturity minus anthesis, for Hist (black), Rcp45 (green), Rcp85 (red).

(apologies for colors, we will redo it)

We note that under Fx2 the season's' length slightly increases; this could be related to model reaching in simulations more frequent physical conditions of "too slow grain filling".

[Figure]

(supplement OK). Also, what was the method for the correlation analysis?

For correlations we used least square fitting method.

Correlations of Harvest with precipitation, (for models, for the two scenarios and Hist and treatment) are now shown in a new supplementary: S2.

31. L319: "H difference Hist minus scenario"? Was reformulated

32. Fig. 8: Needs in-figure legend explaining the colors. From the GMD guidelines at

https://www.geoscientific-model-development.net/submission.html#figurestables:

"A legend should clarify all symbols used and should appear in the figure itself,

rather than verbal explanations in the captions."  corrected in Figures in this version

**33. L337-479 (Sect. 3.3):**

**f. Instead of GX and GI, refer to these percentile ranges as "upper"/"top"**

**and "middle"/"intermediate".**

The suggestion was implemented; we used top/intermediate

**Also, why is the intermediate range 25th-**

**70th (asymmetric around median) rather than 25th-75th?**

We use now the interval 25% - 75

**g. Again, avoid the use of things like Pi and P0i, which make this section**

**hard to parse. Use words instead.** We used "parameters"

34. L372: Why are some numbers in parentheses? corrected

**35. L380-385: I don't understand this almost at all.** Was rephrased

36. Fig. 9:

h. Legends should have sowing date + fertilization level instead of TR#.

For all legends we implemented your suggestion

i. What is ORD? Removed now

j. All the text about Fig. 9 refers to percentile ranges, so those should be the X; axis are computed now as percentile ranges

axis, not rank. Specifically be sure to mark the 2.5th, 25th, and 70th percentiles,

labeling ranges GI and GX.; the 2.5th, 25th, and 75th are located

k. Each one of these lines is an ensemble across three climate models, right?

What is the inter-model variation like? (we added this information in Supplement 3)

l. Fig. 9a: What is the arrow? (removed, we use now build-in rectangle to point the aread discussed in the text)

m. Why are lines in Figs. 9b and 9c not monotonically increasing?

The slopes of response are different in function of treatment, the curves intersect hence, in the differences fields, this results in non-monotonic response

37. L415-417: Please include P# labels here for ease of comparing the text to the figure. Done

38. L418: What are the "main stages of the development"? We now Specified

39. Fig. 10:

**n. Too small.** Fig. 10 was redone

**o. I don't understand what the X axes are supposed to be here.**

Axes were changes to percentiles of change normalised, to allow comparison of percentile of the parameter change, among parameters.

p. Where is Harvest? (Figure was redone)

q. Needs in-figure legend explaining the colors. From the GMD guidelines at

https://www.geoscientific-modeldevelopment. (requirement applied)

net/submission.html#figurestables: "A legend should clarify all

symbols used and should appear in the figure itself, rather than verbal

explanations in the captions."

r. What are the things in the background? Full ensemble ranges for red and

black lines? Why not also blue? (dots are now explained in the caption, as well blue omitting for clarity in the figure: RCP4.% is intermediat to Hist and RCP8.5 in all cases, so was shown only its running mean)

40. Fig. 11: Fig.11 was redone

**s. Too small.**

**t. I don't understand what the X axes are supposed to be here.**

Axis were transformed to show percentiles of the change in the parameter

u. Where is Harvest? (percentiles shown in the new figures)

v. Needs in-figure legend explaining the colors. From the GMD guidelines at

https://www.geoscientific-modeldevelopment.

net/submission.html#figurestables: "A legend should clarify all

symbols used and should appear in the figure itself, rather than verbal

explanations in the captions." we aligned with the requirement

**41. L 462-469: I don't understand this at all.**

"Annex" (should be "Appendix" in *GMD*'s style):

42. Please number the steps. done

43. L805: Repeat starting from which step? Was now specified

44. Consider putting this in Sect. 2 (Data and Methods), because that section is rather

short anyway, and *GMD* encourages technical details. The suggestion was followed

Supplement:

**45. All figures: Do not use red and green in the same figure, as this is dijicult for**

**people with the most common form of color-blindness. See yellow box at the**

**top of https://www.geoscientific-modeldevelopment.**

**net/submission.html#figurestables (removed red-green)**

46. Fig. S1:

**a. Move back to main text (see above). Requirement followed**

**b. Use date + fertilization instead of TRT #. Requirement followed**

c. What are the four-digit numbers? The observed values? Why include these?

d. Many colors are hard to see against the white background. The Figure was redone

e. Missing values should be represented as breaks in the lines rather than zero.

**47. Sect. S2: Just delete this; see comment about L110 above. Supl Section S2 was deleted**

48. Fig. S3:

**f. Too small.**

g. Why here do you split into 1-200 and 201-1890 as opposed to the percentile (plot is a running mean, now pointed in caption)

ranges from the main text?

**49. Fig. S4 is so small, and the image quality is so low, that the figure is**

**unintelligible. (kept only the main results, parameters P1 and P3 for clarity)**

---

## Author Response (AR3)

**Dear Editor, dear Reviewers**

We thank you very much and are indeed, highly grateful for your support, valuable suggestions and corrections to the manuscript.

Here are the answers point-by-point to the required revision.

**Re-review 2: "A modeling System for Identification of Maize Ideotypes, optimal sowing dates and nitrogen fertilization under climate change -PREPCLIM-v1" (gmd-2024-105)**

The authors have significantly improved this manuscript, but some issues still remain. I think the editor can handle them, though, so I'm happy to say accept after minor revisions.

General

1) Thank you for adding the link to the info platform; I was able to access it successfully. (That's a very nice interface!) Please also add the link to the Code Availability section.

A1: added to the Code Availability section

2) Thank you for adding code/data DOIs. Please also mention them in the Code and Data Availability sections.

A2: added to the Code Availability section

3) Figures are still too low-resolution, and this results in some of them being hard to read. The authors mention that they're now 1000x800 pixels, but that by itself doesn't mean anything—what matters is the pixels *per inch*. The GMD submission guidelines specify at least 300 ppi. Fig. 8, for example, is conservatively about 6 inches wide.

That would require it to be at least 300 x 6 = 1800 pixels across, not 1000. Note that using PDF instead of PNG would be preferable for most of these plots, as—being a vector-based format—it allows "infinite resolution." Fig. 8 might be an exception, because the large number of objects (background points) would cause the PDF to be extremely large.

A3: We have redone all the figures of the text and of the Supplement by doubling the initial resolution, we used now x2000; y1600 png.

Sect. 2: Data and Methods

4) New text describing P2 in the tracked-changes version (L150-152) seems not to have made it into the final manuscript. Was this intentional? I find the sentence after "Or:" to be a helpful description.

A4: We included this sentence in this revised form at lines 143-144 in the new Track-changes version.

Sect. 3: Results

5) L280: Replace "exper" with "experiment".

A5: This is now corrected (line 282 in the new Track-changes version)

6) L294-297 (Fig. 4 caption):

a) Replace "exper" with "experiment".

A6a: This was corrected (at line 328 in the new Track-changes version)

b) Mention that the lines represent the mean for each treatment x climate.

A6b: This mention was include at line 330 in the new Track-changes version.

7) L286-291: "Under warmer climates we note more frequent occurrences of critical situations with suboptimal grain filling and potential crop failure, under fertilization....

In our study premature ending of simulated vegetation season occurred more frequently in treatments with higher nitrogen fertilization, leading in average only small changes in maturity days." Reiterating my request that this be illustrated with a figure (in the Supplement is okay). The authors provided one in their response (albeit without a legend or Y-axis labels) but seemingly not in the manuscript or Supplement.

A7: We included the Figure specified by the reviewer, in the Supplementary (the new S3), and included the legend of Y-axis labels. S3 is referred at line 293 in the new Track-changes version.

8) L303: "in the Ctrl and in model simulations": Replace "model" with "future". They're all *model* simulations.

A8: "Model" was replaced with "future" at line 336 (of the in the new Track-changes version)

9) L305-308: When I first re-read this, I thought "they should really look at the changes in extremes, too"—which you do later. Consider saying here something like "Further analysis on the change in intermediate and extreme harvest values can be found in Sect. 3.3.1," or reorganizing to put these analyses together.

A9: We included this note at lines 343 and 344 of the new Track-changes version)

10) L311-313: "The correlation along sowing dates between H and accumulated precipitation until maturity (Pmat, Fig.6), is r(H, Pmat) >0.96 in both scenarios." Fig. 6 does suggest this, but showing the dots for each ensemble member would help. It would match the text I quoted even better to draw the best-fit lines for each point cloud rather than a line through their means at each date. Also, the X-axis (and independent variable in the correlation tests) should be the quantitative sowing date rather than the categorical "treatment"—these are almost identically-spaced but not exactly.

A10: We have redone these Figures (Fig.4, Fig5, Fig6), now these use unequally spaced intervals on Ox such as to represent not the number of the treatment but the number of days between sowing dates. The difference is small but is apparent on the Figures, the dates being: 01 April, 15 April, 01 May, 15 May.

Regarding using best-fit curves instead of ensemble means, for this work this would bring here some new, different issues, difficult here to link to the rest of the results (that target ensemble means), but it is an interesting idea that will be considered in a further investigation.

11) L354: Upper limit of "intermediate" interval still says 70% here, whereas in their response the authors say they changed it to 75%.

A11: This was corrected to 75% at line 417 of the new Track-changes version); all the figures (Fig.7b.c) were containing the correct value, 75% for the definition of the intermediate H values.

12) L355: Replace "projected higher H values in GI" with "projected higher intermediate H

values".

A11: This was replaced at line 418 of the new Track-changes version;

13) L369-375:

a) L371-372: Replace "GI and also in GX" with "both the intermediate and top percentiles".

A13a:  This was replaced at line 436 of the new Track-changes version

b) L374: "leading"? Reformulated at line 440 of the new Track-changes version

A13b:

c) Mention any interesting results from Fig. S3. E.g., how some ensemble members actually show improvements. Also, it seems like there is substantial inter-model spread in the historical and to a lesser extent future periods—this is not a dealbreaker but it should be acknowledged in the main text.

A13c: These were now noticed specifically in the text (lines 415-416 Track-changes) with reference to the two Supplementary materials: to the Fig.S3 (showing the link between precipitation and Harvest) and to the Fig.S5 (showing the inter-model spread in simulated Harvest).

14) Fig. 7a

a) This is actually harder to read in some ways than it was before (then Fig. 9a). The first two lines for each color are impossible to distinguish. Using a PDF as requested in the guidelines could help with this.

A14a: We changes the thickens of the lines and also changed the line style of the lines (for first two sowing dates) to make it more distinct, in Fig.7.

b) Fig. S3 should be mentioned in the Fig. 7a caption. This was done in the caption of Fig.7a, line 487 (new Track-changes text)

c) I still don't understand what I'm supposed to take away from the "mitigation window." How was it drawn, exactly? I.e., how were the edges determined?

A14c: The square is defined by the points where H-percentile curves obtained in different management scenarios (as a function of genotype), are first crossing each-other. This intersection is due to different curvatures of H(genotype) functions for the various treatments (showing a different response to genotype perturbation under different managements).

These squares delimit genotype parameter-areas where, using a same (or close-range) genotype and just changing the management (sowing or fertilization) one can obtain the same or even an improved H, for a given climate.

This is important feature that can be used in practice, because one can reduce pollution (fertisation) but using the alternate management (sowing date or genotype) using this identified parameter-area, called "mitigation window"; hence can mitigate the climate change change effects without loosing Harvest potential (discussed in 3.3.1 ii, window definition explanation added now in the caption of Fig.7a).

15) Figs. 7b and 7c: Y-axis labels should be "Difference in harvest relative to Hist".

A15: We modified in the figure the Y-axis labels, to: "Difference in harvest relative to Hist".

16) Fig. 8 is still too low-resolution to properly display the point clouds in the background.

A16: we enhanced the resolution for this Figure (as for all) to x2000 y1600

17) L423: "Ox" should be changed to "X axis".

A17: done

18) L452: "Ox" should be changed to "X axis".

A18:This was replaced at line 541 and line 586 (new Track-changes text)

19) L459: "lead" should be "lead to".

A19: Was corrected at line 593

20) L462-466: I still don't understand this part (although the preceding part of this section is better than before; my previous comment 41). I don't really get what "expectancy" means, and I don't know what the Y-axis in Fig. S5 is. I think "border" at L464 should be "broader"?

A20: replaces "expectancy" with "probability"

A20: corrected "border" to "broader"

Sect. 4: Discussions

21) L500: Leftover "P0i" that didn't get changed.

A21: Replaced at line 634 with "genotype parameter value"

22) L544: Leftover "P01" that didn't get changed.  Line 678

A22: Replaced at line 678 with "genotype parameters"

23) L545: Leftover "P0i" that didn't get changed.

A23:Replaced at line 679 with "these cultivar dependent parameters"

Supplement:

24) Throughout: Start section headers with "Section" and figure captions with "Figure" to clearly distinguish them each other.

A24: This was done in Supplementary material: Supplementary materials are named "Section" and figures are Fig.S* (in the same way referenced in the text)

25) Fig. S.1b (previously Fig. 5): The authors mention that my comments 24b ("Needs in figure legend explaining the lines, their colors, and what the shading represents") and 24c ("Y-axis labels needed with text explanations and units") were addressed, but it doesn't look like those changes made it into the updated Supplement file.

A25: We addressed these in the new Supplementary material version.  All contain now legend for lines, colors and shading and explained axis.

26) L19: "Fertlization" typo

A26: Done

27) L26: "string" should be "strong".

A27: Done

28) L28: "dt-line" should be "dot-dash line".

A28:  is now corrected

29) Fig. S3:

a) Refer back to Fig. 7a in the caption.

A29a: This reference was included in the caption of Fig.7a (line 487)

b) These figures should also be remade to match the design of Fig. 7a.

A29b: These figures were remade, with same pattern as Fig.7

30) Fig. S5:

a) What is the Y-axis here? Now this is Fig.S7, and the

A30a: we added a caption line to explain Y-axis in this Figure, now Fig.S7.

b) Why split at the 200th parameter instead of percentiles as in the rest of the paper?

A30b: The split was done at the point (range) of maximal change in the slopes in Scenarios relative to the slopes in Hist, in order to identify the percentile range most affected by climate. The maximal change is at about 10-20% percentile in all cases (coherent with Fig.7b,c), such as slopes higher in Scenarios in the first interval become lower afterwards compared to Hist, in function of the genotype parameter. The relative change is slope indicates how (and how much) to perturb a parameter to maintain about current H percentile.

c) Linear regression doesn't look like a good fit for the two plots on the left.

A30c: Yes, it appears for some parameters that the linear fit is poorer, mainly for un-fertilised case (Fig.S7)

However, here we are interested first on the sign of the relative change and then (also interesting) on how this difference evolves (where it accelerates or is more linear as a function of the parameter).

31) L59: Replace "TR12" with the actual details of the treatment.

A31: done (the treatment is defines as in figures, by "date_fetilization" and reference is to GTR, Table 1b)

32) L60: Delete "GI".

A32: done

---

## Author Response (AR4)

**Dear Editor,**

We thank you very much and are highly grateful for your support, valuable suggestions and corrections to the manuscript.

Here are the answers point-by-point to the required revision.

- Please replace all occurrences of "expectancy" with "probability".

We replaced " expectancy" with "probability" in all occurrences. In the new text at lines:  383; 383; 393; 515; 519; 589; 590

- Fig. S1b: Please add explanation of the numbers on the figure (R? R-squared?), as well as the line colors.

In the Figure S1b (line 20) we added the explanation of numbers, we clarified the colors inside the plot-box and we clarified the Figure name as the notation R10 was not explained.

We used R10 for the Extreme climate indicator: "number of days with precipitation > 10mm", but this was not enough explained. Now, we put a more clear Figure Title, with a more explicit description of the indicator" "Heavy precipitation days", and the dekade of accumulation (cf. Y-axis). In FigS1b caption, this clarification was added accordingly at line 41.

- Fig. S3: Please refer back to Fig. 7a in the caption of Fig. S3.

The link between Fig7a and Supplementary changed because following Reviewers, new Supplementary figures were included at last review (initially 4, then since last revision there are7). Hence the Supplement S3 to which the Referee requires this specification became at the last revision, S5. In the Caption of S5 the Fig. 7a is mentioned at line 166: " Harvest time mean simulated by models: ECEARTH-ICHEC (left), MPI (Middle) and CNRM (right). As in Fig.7a (that shows the same but for the ensemble mean), are shown percentiles of the H distribution ordered from maximum H values (left) to minimum H (right), logarithmic scale."

In the Fig.7a,  the S5 figure is also mentioned at line 417: ".. see also the models H distribution in Fig.S5"

**Other small corrections:**

We checked the full text and supplementary and some other small corrections were done, seen in the Track changes document.

Most of these are typo errors, (grammar, numbers of figures, caption clarifications), or linking words for a more cursive reading.

There are few changes that do not fall in these categories and are described here:

- Figure 3 (line 273): we thought could be good to add Y-axis label, in order to have axis description as the other Figures

- Figure 7 (in the top line, when grouping Figure panels, for the P4 figure the labels were hidden): we re-done the grouping of panels, at line 455

- Lines 500-505 were doubled (as content), so one version was removed; the statements are maintained at the same place (lines 477-478 and 486-488).

-   in the Abstract we removed the line: "Soil initial conditions were found to significantly influence yield responses".

The statement is correct, but the figures that were in the initial text showing the experiments that emphasized this, were suggested by Reviewers to be removed as it was spreading too much the topic. We fully agreed and removed this part, but not from the Abstract at that time. Now we proposed to remove it from the Abstract as it has no correspondent in the text;

- at the Conclusions we were missing a statement that answers to question iii) presented at line:  585: "Can be genotyping a (better) solution for adaptation under climate change in the region?"

However this aspect was analyzed in the text at lines 382-388; 392-402.

Hence a line of this conclusion was added at line 602: " These results show that genetic approaches offer adaptation strategy support in helping plants to resist drought stress under warming climate, while a projected narrowing of the agro-management options for maintaining a high yield level is emphasised under warmer and drier climate."